# Macro Placement
# by Wire-Mask-Guided Black-Box Optimization

**Yunqi Shi, Ke Xue, Lei Song, Chao Qian**[*]

National Key Laboratory for Novel Software Technology, Nanjing University, China
School of Artificial Intelligence, Nanjing University, China
`{shiyq, xuek, songl, qianc}@lamda.nju.edu.cn`

## Abstract

The development of very large-scale integration (VLSI) technology has posed new challenges for electronic design automation (EDA) techniques in chip floorplanning. During this process, macro placement is an important subproblem, which tries to determine the positions of all macros with the aim of minimizing half-perimeter wirelength (HPWL) and avoiding overlapping. Previous methods include packing-based, analytical and reinforcement learning methods. In this paper, we propose a new black-box optimization (BBO) framework (called WireMask-BBO) for macro placement, by using a wire-mask-guided greedy procedure for objective evaluation. Equipped with different BBO algorithms, WireMask-BBO empirically achieves significant improvements over previous methods, i.e., achieves significantly shorter HPWL by using much less time. Furthermore, it can fine-tune existing placements by treating them as initial solutions, which can bring up to 50% improvement in HPWL. WireMask-BBO has the potential to significantly improve the quality and efficiency of chip floorplanning, which makes it appealing to researchers and practitioners in EDA and will also promote the application of BBO. Our code is available at `https://github.com/lamda-bbo/WireMask-BBO`.

## 1 Introduction

EDA techniques have been widely employed to assist engineers in designing chips [30, 31], while the rapid advancement of VLSI technology has led to an exponential growth in chip scale, posing significant challenges. Particularly, for the important chip floorplanning stage in EDA, which strives to optimize power, performance, and area metrics while adhering to constraints such as congestion and density [45], the number of cells to be placed on the chip canvas increases rapidly and their routing relationships also become more complex, requiring innovative and efficient methods.

During the floorplanning placement stage, the position of each cell on the chip canvas is established. A modern chip typically comprises thousands of macros (i.e., individual building blocks such as memories) and millions of standard cells (i.e., smaller basic components like logic gates). The designer provides a netlist, which outlines the design requirements and serves as a large-scale hyper-graph containing numerous hyper-edges (also called nets) that represent the routing relationships among macros and standard cells. Traditionally, the placement problem is divided into two successive stages [4]: macro placement, which is usually addressed by heuristic or learning methods [42, 44], and standard cell placement, which is usually addressed by analytical solvers [13, 28]. After placing all the cells (including macros and standard cells), a routing stage is performed. The general flow of chip floorplanning and routing is illustrated in Figure 1. In this work, we concentrate on macro placement due to its greater impact on placement quality: Macros are often larger and more crucial for achieving optimal placement [4, 47].

---

[*]Corresponding Author

37th Conference on Neural Information Processing Systems (NeurIPS 2023).

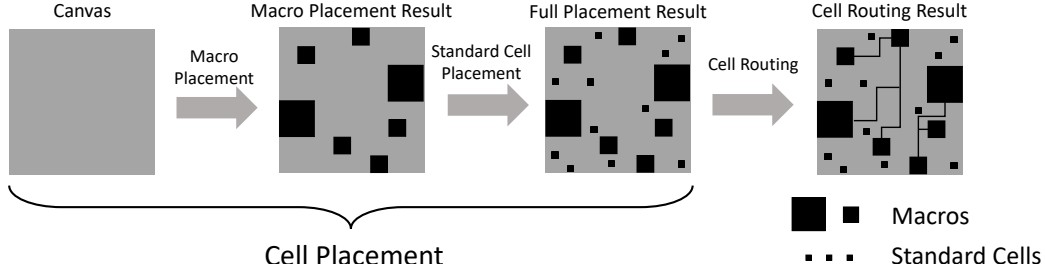

Figure 1: General flow of chip floorplanning and routing, where the routing stage basically relies on the placement result. We focus on the macro placement task in this paper.

Earlier methods often formulate macro placement as a rectangular packing problem, where solutions are represented by sequence pair (SP) [33], $B^*$-tree [11], corner block list (CBL) [19] or other data structures [34, 27], and solve it by simulated annealing (SA) [1, 18, 24, 40]. This kind of method suffers from the poor scalability due to the quadratic complexity of decoding a genotype solution to a phenotype placement. Analytical methods [12, 13, 28, 29] place macros and standard cells simultaneously, and relax the task to a mathematical programming problem, which can be solved efficiently. However, they cannot guarantee the non-overlapping between cells, which is a hard constraint of cell placement. More recently, by dividing the chip canvas into discrete grids and formulating the task of placing macros onto grids step by step as a Markov decision process (MDP), reinforcement learning (RL) methods have been applied, showing promising performance [14, 15, 26, 32]. But their fast convergence (after only a few hundred evaluations) observed in experiments may imply that the huge search space of placement is still underexplored, and further improvement is expected. A detailed review of these existing methods is provided in Section 2.

In this paper, we propose a new BBO framework for macro placement. To allow better exploration, a solution is directly represented by the coordinates of all macros on the chip canvas. HPWL is used as the minimization objective function. The key of our framework is that in order to improve efficiency, we use an elaborate evaluation procedure, which greedily improves a solution (with the goal of minimizing HPWL while avoiding overlapping) before evaluation. Concretely, using the wire mask [26] to record the increment of HPWL by moving a macro to each candidate grid on the chip canvas, the macros in the solution are sequentially adjusted to the nearest best grid. This framework is briefly called WireMask-BBO, which can use any BBO algorithm to solve the resulting problem.

Experimental results on multiple popular benchmarks demonstrate that WireMask-BBO, when equipped with different BBO algorithms such as random search (RS), Bayesian optimization (BO), and evolutionary algorithms (EA), significantly outperforms the compared representative methods, including traditional BBO methods based on packing formulation, analytical methods, and RL methods. Especially, WireMask-EA (i.e., WireMask-BBO equipped with EA) generates the highest-quality placement in 6 out of 7 different chip benchmarks, and surpasses the state-of-the-art MaskPlace [26] in 8 minutes on average. Besides, as an optimization-based method, WireMask-BBO can fine-tune any existing placement, regardless of how it was generated. That is, WireMask-BBO can be combined with any existing macro placement method as post-processing. Our experiments show that such fine-tuning can lead to an improvement of up to 50% in the objective HPWL.

Our main contribution is introducing the general framework WireMask-BBO, while not developing new BBO algorithms. In fact, our experiments show that even employing simple BBO techniques has led to superior performance over previous methods. We will open-source WireMask-BBO and use it as an optimization benchmark to encourage the invention of more efficient BBO algorithms for solving macro placement problems, as well as broaden the application scenario of BBO.

## 2   Background

In this section, we introduce the macro placement problem, and the existing methods which can be generally categorized into packing-based, analytical, and grid-based RL methods.

## 2.1 Macro Placement

The input for a macro placement problem constitutes a netlist $H = (V, E)$, where $V$ denotes the information (i.e., height and width) about all cells designated for placement on the chip, and $E$ is a hyper-graph comprised of hyper-edges $e_i \in E$, which encompasses multiple cells (including both macros and standard cells) and signifies their interconnectivity during the routing phase. A macro placement solution consists of the positions of all macros $\{v_i\}_{i=1}^{k}$ with the coordinates of each macro $v_i \in V$ on the chip canvas expressed as $(x_i, y_i)$, where $k$ denotes the total number of marcros. To facilitate a comprehensive understanding of the macro placement problem, we present several key metrics for evaluating placement outcomes.

**HPWL** is the predominant metric for gauging placement quality, as it offers a precise estimation of the wirelength necessary for routing [9, 22, 38]. A shorter wirelength decreases delay and power consumption, enhancing overall performance [36]. To calculate HPWL of a macro placement solution $s$, each hyper-edge $e_j \in E$ corresponds to a rectangle area characterized by its lower-left endpoint, with coordinates $(\min_{v_i \in e_j} x_i, \min_{v_i \in e_j} y_i)$, and its upper-right endpoint, with coordinates $(\max_{v_i \in e_j} x_i, \max_{v_i \in e_j} y_i)$. That is, the rectangle is the smallest one bounding all cells within $e_j$. Let $w_j = \max_{v_i \in e_j} x_i - \min_{v_i \in e_j} x_i$ and $h_j = \max_{v_i \in e_j} y_i - \min_{v_i \in e_j} y_i$ denote the width and height of the rectangle, respectively. Then,

$$\text{HPWL}(s, H) = \sum_{e_j \in E} (w_j + h_j). \tag{1}$$

**Congestion** serves as a vital metric in determining the routability of a given placement, exerting a direct influence on the manufacturing process. A widely-adopted approach for approximating congestion is rectangular uniform wire density (RUDY) [41]. The congestion of a grid on the canvas can be represented as the cumulative impact of all hyper-edge rectangle areas encompassing the grid. By selecting the top $10\%$ congested grids (denoted as $G^*$) and computing the mean of their congestion, the RUDY value of a macro placement solution is calculated as $\frac{1}{|G^*|} \sum_{g_i \in G^*} \sum_{e_j \in E(g_i)} \frac{w_j + h_j}{w_j \cdot h_j}$, where $|G^*|$ is the size of $G^*$, $E(g_i)$ denotes the set of hyper-edges whose rectangle areas cover $g_i$, and $(w_j + h_j)/(w_j \cdot h_j)$ measures the impact of the rectangle area of hyper-edge $e_j$.

**Density** measures the overlap degree between cells by employing an electrostatic analogy [29]. It is often treated as a penalty to diminish overlap, yielding a more uniform distribution of cells across the chip area [4, 13, 28, 29]. Further discussion will not be provided here, as our formulation ensures non-overlapping, which is a hard constraint of macro placement.

**Area** denotes the area of the minimum bounding rectangle that encloses all the macros. It was prevalent when the focus was on tightly packing macros and minimizing the area they occupied [10, 42]. However, when considering fixed-area chips as in [15, 26, 28, 32] and our work, the placement of cells optimized is to reduce wirelength and congestion rather than minimize the area.

## 2.2 Packing-based Methods

As the focus was once on minimizing the bounding area of given macros, which can lead to a higher chip area utilization ratio [31], the packing formulation emerged as a natural and straightforward approach for the macro placement problem. Each macro, represented as a rectangle, has to be packed within a specified chip canvas, with the objective being to optimize the weighted sum of the area metric and HPWL metric. Several solution representation methods have been proposed, such as SP [33], B*-tree [11], CBL [19], etc. These genotype solutions will be mapped to concrete macro placement (phenotype) solutions for evaluation. SA is often employed [1, 18, 24, 40] to solve such BBO problems by perturbing the genotype to generate new offspring solutions and evaluating the corresponding phenotype to determine whether to accept the perturbation.

To handle both macros and standard cells, a divide-and-conquer idea is introduced [39, 43]. The standard cells are first clustered into blocks using either the logical hierarchy or min-cut-based partitioning algorithms [23, 37]. Placement is then performed on the mix of macro blocks and clustered standard cell blocks, often by SA using packing formulation [1, 2]. Finally, the standard cells are re-allocated by detailed placement. While reducing the problem size, clustering standard cells into rectangular blocks may cut some connections and hinder finding an optimal solution.

Recently, RL has been introduced to decide which perturbation operator should be selected and which macro should be perturbed by SA [46]. Moreover, [5] formulates the CBL establishment as a

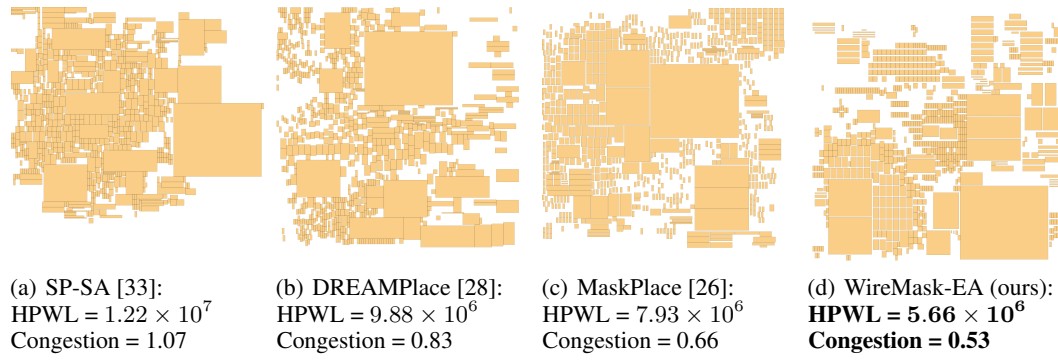

(a) SP-SA [33]:
HPWL = $1.22 \times 10^7$
Congestion = 1.07

(b) DREAMPlace [28]:
HPWL = $9.88 \times 10^6$
Congestion = 0.83

(c) MaskPlace [26]:
HPWL = $7.93 \times 10^6$
Congestion = 0.66

(d) WireMask-EA (ours):
**HPWL = $\mathbf{5.66 \times 10^6}$**
**Congestion = 0.53**

Figure 2: Visualization of macro placement on the circuit benchmark *adaptec3* using the proposed WireMask-EA and three representative methods. The macros are represented by yellow rectangles. The performance is evaluated in terms of HPWL and congestion; the lower the value, the better.

MDP and constructs a CBL genotype step by step. These advancements continue to grapple with the drawbacks of packing, such as poor scalability, which cannot be solved fundamentally because the drawbacks come from the quadratic complexity of mapping from genotype to phenotype [33]. Besides, at least one edge of each macro under the packing setting must be attached to another macro or the canvas's edge, making the placement quite congested, as visualized in Figure 2(a). Note that the congested macro placement will limit the space for subsequent standard cell placement, and lead to the bad performance of full placement, which will be shown in Table 3.

## 2.3 Analytical Methods

Analytical methods [12, 13, 28, 29] place macros and standard cells simultaneously, and relax the full placement task to a mathematical programming problem, which can be solved efficiently. For example, DREAMPlace [28], a state-of-the-art and highly efficient analytical placement method, recasts the full placement task as $\min \mathrm{WA}(\boldsymbol{s}, H) + \lambda \cdot \mathrm{Density}(\boldsymbol{s}, H)$, where WA represents the smooth weighted-average wirelength (originally proposed by [20]) for approximating HPWL, Density denotes a differentiable density metric for penalizing overlapping, and $\lambda$ is a trade-off factor. This problem is then solved numerically using classical mathematical optimization techniques (e.g., gradient descent), to rapidly generate high-quality full placement. Figure 2(b) shows a macro placement generated by DREAMPlace, which is better than that generated by packing in Figure 2(a). However, analytical methods cannot guarantee the non-overlapping between cells. Even employing macro placement legalization techniques, numerous overlaps may persist [4].

## 2.4 Grid-based RL Methods

As the number of macros increases, packing-based methods encounter scalability challenges, while analytical methods may produce overlapping placements that are impractical for manufacturing [4]. To meet the demands of modern chip design and fully leverage RL, the Graph placement published in *Nature* [32] divides the chip canvas into discrete grids, with each macro assigned discrete coordinates of grids, and formulates the placement problem as a MDP, wherein the agent decides the placement of the next macro at each step. Notably, no reward is given until the final step, when all macros are placed. The ultimate reward is the weighted sum of HPWL and congestion. Compared with packing-based methods, the grid-based design eliminates edge attachment and provides room for standard cell placement while offering improved scalability. The length of the MDP grows linearly with the number of macros, as opposed to the quadratic complexity of packing formulation [33]. DeepPR [15] integrates convolutional and graph neural networks during the embedding stage, and introduces an intrinsic reward to promote exploration. However, DeepPR brings overlaps. Though PRNet [14] incorporates the overlap area into the reward function as a penalty, the non-overlapping issue still exists as observed in their experiments.

A key limitation of the above-mentioned methods is the absence of extrinsic rewards until the final step, leading to sub-optimal performance for a long MDP with over one thousand steps. To

address this issue, MaskPlace [26] introduces a dense reward RL pipeline, incorporating a view mask for gathering global information, a position mask to ensure non-overlapping, and a wire mask to evaluate the placement of the current macro. Notably, the wire mask offers an immediate reward, calculated as the increase in HPWL after placing the current macro. These three masks are processed by a convolutional neural network and serve as input features for both the value and policy networks. MaskPlace can generate high-quality non-overlapping placement results in an affordable time. Figure 2(c) gives a placement example, which is better than that generated by packing and analytical methods as shown in Figures 2(a) and 2(b). From the experiments in [26], we can also observe that MaskPlace converges after only a few hundred evaluations, implying that the huge search space of placement may be still underexplored. The experimental results in Section 4 will confirm this conjecture, showing that our proposed framework will bring significant improvement.

## 3 Proposed Framework WireMask-BBO

This section is devoted to our proposed WireMask-BBO, where Section 3.1 introduces the problem formulation and Section 3.2 is concerned with optimization methods to solve the resulting problem.

### 3.1 Wire-Mask-Guided Problem Formulation

**Solution representation.** To allow better exploration, a macro placement solution $s$ is directly represented by the coordinates of all macros $\{v_i\}_{i=1}^k$, i.e., $s = (x_1, y_1, \ldots, x_k, y_k)$, where $(x_i, y_i)$ denotes the coordinates of the macro $v_i$ on the chip canvas. For example, there are three macros A, B and C for the placement task in Figure 3, and thus a solution $s$ is represented by $(x_1, y_1, x_2, y_2, x_3, y_3)$.

**Objective evaluation.** The metric HPWL in Eq. (1) is used as the objective function to be minimized. But if optimizing in the solution space directly, it is difficult to efficiently find a solution that has a small HPWL value and satisfies the non-overlapping constraint. To improve the efficiency, a greedy improvement strategy is applied to a solution before evaluating it.

As in [32], the chip canvas is first divided into discrete grids. In the process of improving a solution greedily, each macro is moved to a grid by letting the macro's bottom-left corner situate at the bottom-left corner of the grid. The order of adjusting the position of a macro $v_i$ is determined by the area of all the cells connected with $v_i$, i.e., all the cells in the hyper-edges containing $v_i$. Note that each hyper-edge contains a set of connected cells. The positions of all macros will be adjusted sequentially in the decreasing order of their corresponding computed areas, because a macro with a larger computed area implies connecting with more large cells and thus is intuitively more important.

Let $v_1^*, \ldots, v_k^*$ denote the ordered macros. Assume that the positions of $v_1^*, \ldots, v_{i-1}^*$ have been adjusted. When considering $v_i^*$, a wire mask $W_i$ introduced in [26] is first computed, which records the increase of HPWL by placing $v_i^*$ to each candidate grid, given $v_1^*, \ldots, v_{i-1}^*$ already adjusted. Note that those grids that will lead to overlap or exceeding the canvas boundary after placing $v_i^*$ are excluded. Then, $v_i^*$ is moved to the grid with the least increment of HPWL. If such a grid is not unique, $v_i^*$ is moved to the nearest one among them. This is actually a greedy step that moves $v_i^*$ to the grid with the least increment on HPWL given the placement of $v_1^*, \ldots, v_{i-1}^*$. This process is repeated until the positions of all the macros have been adjusted. The HPWL of the resulting solution with all the adjusted macros is treated as the objective value of the original solution. Thus, a solution and its improved version in objective evaluation can be viewed as the genotype and phenotype representation of a macro placement. The detailed process of objective evalution is presented in Algorithm 1.

Figure 3 gives an example illustration of objective evaluation. For the input netlist $H = (V, E)$, $V$ contains three macros A, B and C, and the hyper-graph $E$ contains two hyper-edges: one encompasses macros A and B, and the other encompasses A and C. Each hyper-edge signifies the interconnectivity of the macros contained by it. The chip canvas is partitioned into $5 \times 5$ grids. Assume that the width and height of each grid is both 1. The solution to be evaluated (as shown in Figure 3(a)) consists of the positions of macros A, B and C. According to the hyper-graph $E$, we know that during the routing phase, macro A is connected with B and C, while macro B and C are both connected with only A. Thus, the corresponding areas computed in line 1 of Algorithm 1 are decreasingly ordered as A, B and C, which is just the order in line 2 to adjust the positions of macros sequentially.

When trying to adjust the position of macro A, a wire mask is first computed as in line 5 of Algorithm 1, which is shown in Figure 3(b), where the number in each grid is the increment of HPWL

**Algorithm 1** Objective evaluation of WireMask-BBO

**Input**: solution $s$, netlist $H = (V, E)$, and number $m$ of partitions
**Output**: improved solution $s'$, and the corresponding objective (i.e., HPWL) value
**Process**:

1: For each macro $v_i \in V$, compute the area of all the cells connected with it, i.e., $\cup_{v_i \in e_j \in E} e_j$;
2: Order all macros decreasingly, denoted as $v_1^*, \ldots, v_k^*$, according to their corresponding areas;
3: Initialize the canvas as $m \times m$ grids, and let $f = 0$;
4: **for** each macro $v_i^*$ **do**
5:     Generate wire mask $W_i$ according to the updated positions of $v_1^*, \ldots, v_{i-1}^*$ (see Algorithm 1 in [26]), which records the increment of HPWL by moving $v_i^*$ to each candidate grid;
6:     $Q \leftarrow$ the set of grids that has the minimum increment of HPWL (denoted as $f_i$) in $W_i$;
7:     Select the grid $g$ from $Q$, which has the smallest Euclidean distance to the macro $v_i^*$;
8:     Update the position of $v_i^*$ to be that of the grid $g$;
9:     $f \leftarrow f + f_i$
10: **end for**
11: $s' \leftarrow$ the updated positions of all the macros
12: **return** $s', f$

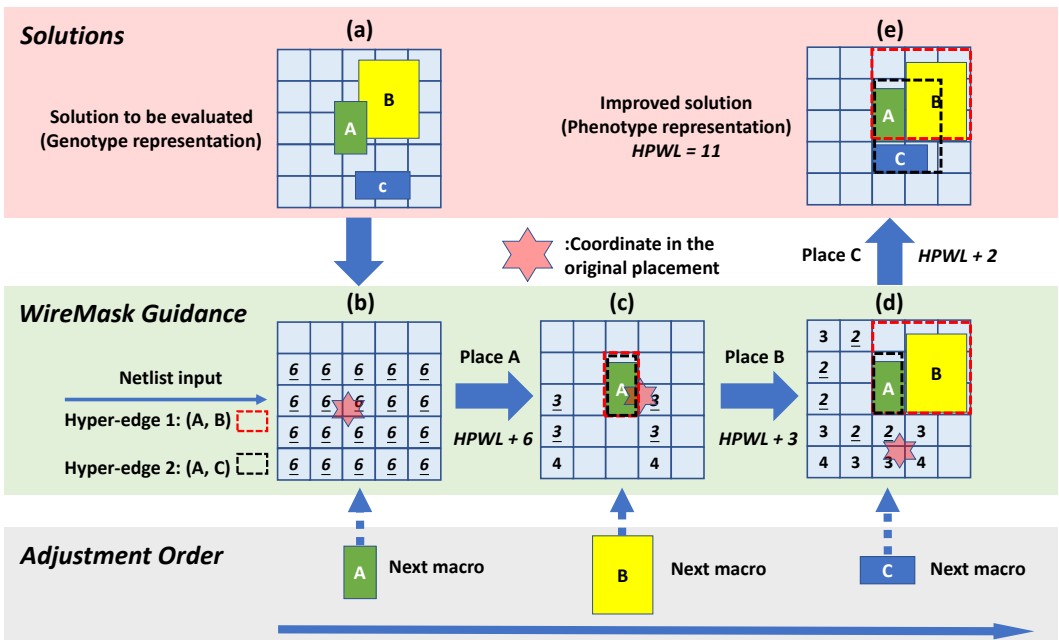

Figure 3: An example illustration of objective evaluation of WireMask-BBO.

led by placing macro A in that grid. For example, if placing macro A in the center grid as shown in Figure 3(c), both the two hyper-edges correspond to the smallest rectangle bounding macro A (i.e., the red and black dotted square in Figure 3(c)) for computing HPWL in Eq. (1), and thus the corresponding HPWL is 6, which is the sum of half perimeters of these two rectangles. As the initial HPWL before placing macro A is 0, the increment of HPWL is 6. Note that when adjusting the position of a macro, its bottom-left corner must be situated at the bottom-left corner of a grid, and grids without numbers in the wire mask indicate that the current macro should not be placed there due to overlap or exceeding the canvas boundary. The underlined italic number in the wire mask denotes the least HPWL increment, representing the locally optimal choice, i.e., the set $Q$ in line 6 of Algorithm 1. Then, as shown in lines 7–8 of Algorithm 1, macro A is placed at the local optimal grid, which has the smallest Euclidean distance to it. The coordinate of macro A in the original placement is denoted by the red star in Figure 3(b), and thus it is adjusted to be situated at the grid closest to it, as shown in Figure 3(c).

The positions of macros B and C are adjusted similarly. Figure 3(c) shows the adjusted placement of macro A, and also the wire mask when trying to place macro B. Figure 3(d) shows the adjusted placement of macros A and B, and also the wire mask when trying to place macro C. Figure 3(e) gives the final improved placement after adjusting the positions of all macros. As HPWL is the sum of all hyper-edges' bounding rectangles' half perimeters, these rectangles can be updated each time a macro is placed, as shown by the red and black dotted squares in Figure 3. That is, for any individual macro, it is only necessary to update the bounding rectangles of its associated hyper-edges and compute the increase in their half perimeters. Thus, the calculation of HPWL can be performed incrementally, as in line 9 of Algorithm 1. In Figure 3, the increment of HPWL after placing macros A, B and C is 6, 3 and 2, respectively, leading to the HPWL value 11 of the final improved placement. Note that the red and black dotted bounding rectangles in Figure 3 are shown for illustrative purposes, while the actual HPWL calculation in our experiments is based on pin information.

Therefore, we have formulated macro placement as a black-box optimization problem, where the genotype-phenotype mapping is greedily guided by wire mask. Figures 3(a) and (e) give an example of genotype and phenotype representation. An algorithm for solving this problem will search in the genotype space, where the goodness of a genotype solution is estimated by the objective value of its corresponding phenotype. When an algorithm terminates, the corresponding phenotype of the generated genotype solution will be output as the final macro placement. Note that by the wire-mask-guided mapping process, the phenotype solution is guaranteed to have no overlap, and thus an algorithm can search in the genotype space without considering constraints.

### 3.2  Black-Box Optimization

The above formulated problem can be solved by any BBO algorithm. In our experiments, we employ three simple ones, random search (RS), Bayesian optimization (BO) [17] and evolutionary algorithm (EA) [7]. RS randomly allocates all macros in a solution and evaluates it, recording the historical best. For BO, we adopt the efficient TuRBO approach [16] to directly optimize the continuous coordinates $(x_1, y_1, \ldots, x_k, y_k)$ of all macros, resulting in a dimension of $2k$, where $k$ is the number of macros. For EA, we choose the simple (1+1)-EA [6, 49], which maintains only one solution and iteratively improves it by mutation. We design the mutation operator by randomly selecting two macros and exchanging their coordinates in a solution. Our experiments will show that using these three simple BBO algorithms has been sufficient for the superior performance over previous methods. It is expected to design better BBO algorithms for the proposed formulation of macro placement.

By adopting the grid-based discretization and designing the wire-mask-guided greedy genotype-phenotype mapping, the proposed framework WireMask-BBO can efficiently generate a non-overlapping high-quality placement, addressing the poor scalability issue of packing-based methods as well as the overlapping issue of analytical methods. The use of black-box optimization enhances the exploration ability, making WireMask-BBO able to surpass the state-of-the-art RL-based method MaskPlace [26]. Figure 2(d) visualizes a placement generated by WireMask-BBO equipped with EA, which is better than that by SP-SA (a representative packing-based method) [33], DREAMPlace (a representative analytical method) [28] and MaskPlace [26], as shown in Figures 2(a), 2(b) and 2(c).

## 4  Experiments

We mainly empirically test our method on the ISPD2005 benchmark [35], which was originally proposed as a standard cell placement benchmark with fixed macros. Following conventional practice [15, 14, 26], we modify all macros to be movable for the macro placement problem. The ISPD2005 benchmark contains eight chips, i.e., *adaptec1-4* and *bigblue1-4*. Note that for the *bigblue2* chip, which has over 20,000 macros and 30,000 macro-related hyper-edges, a single objective evaluation by step-by-step placement costs more than an hour. Thus, we do not include this chip in our experiments as [26]. For each chip, the canvas is partitioned into approximately $150 \times 150$ grids heuristically. The detailed statistics (e.g., the number of macros, partitions, etc.) of the chips are provided in Appendix A.

We compare WireMask-BBO with several representative macro placement methods, including the packing-based SP-SA [33], three analytical methods NTUPlace3 [12], RePlace [13], DREAM-Place [28], and three RL-based methods Graph placement [32], DeepPR [15], MaskPlace [26]. As introduced in Section 3.2, we equip the proposed WireMask-BBO framework with RS, BO and EA,

Table 1: HPWL values ($\times 10^5$) obtained by ten compared methods on seven chips. Each result consists of the mean and standard deviation of five runs. The best (smallest) mean value on each chip is bolded. The symbols '+', '−' and '≈' indicate the number of chips where the result is significantly superior to, inferior to, and almost equivalent to WireMask-EA, respectively, according to the Wilcoxon rank-sum test with significance level 0.05.

| Method | Type | adaptec1 | adaptec2 | adaptec3 | adaptec4 | bigblue1 | bigblue3 | bigblue4 ($\times 10^7$) | +/−/≈ | Avg. Rank |
|---|---|---|---|---|---|---|---|---|---|---|
| SP-SA [33] | Packing | 18.84 ± 4.62 | 117.36 ± 8.73 | 115.48 ± 7.56 | 120.03 ± 4.25 | 5.12 ± 1.43 | 164.70 ± 19.55 | 25.49 ± 2.73 | 0/7/0 | 6.86 |
| NTUPlace3 [12] | Analytical | 26.62 | 321.17 | 328.44 | 462.93 | 22.85 | 455.53 | 48.38 | 0/7/0 | 9.00 |
| RePlace [13] | Analytical | 16.19 ± 2.10 | 153.26 ± 29.01 | 111.21 ± 11.69 | 37.64 ± 1.05 | 2.45 ± 0.06 | 119.84 ± 34.43 | 11.80 ± 0.73 | 1/6/0 | 5.28 |
| DREAMPlace [28] | Analytical | 15.81 ± 1.64 | 140.79 ± 26.73 | 121.94 ± 25.05 | **37.41 ± 0.87** | 2.44 ± 0.06 | 107.19 ± 29.91 | 12.29 ± 1.64 | 1/6/0 | 4.86 |
| Graph [32] | RL | 30.10 ± 2.98 | 351.71 ± 38.20 | 358.18 ± 13.95 | 151.42 ± 9.72 | 10.58 ± 1.29 | 357.48 ± 47.83 | 53.35 ± 4.06 | 0/7/0 | 9.00 |
| DeepPR [15] | RL | 19.91 ± 2.13 | 203.51 ± 6.27 | 347.16 ± 4.32 | 311.86 ± 56.74 | 23.33 ± 3.65 | 430.48 ± 12.18 | 68.30 ± 4.44 | 0/7/0 | 8.86 |
| MaskPlace [26] | RL | 6.38 ± 0.35 | 73.75 ± 6.35 | 84.44 ± 3.60 | 79.21 ± 0.65 | 2.39 ± 0.05 | 91.11 ± 7.83 | 11.07 ± 0.90 | 0/7/0 | 4.28 |
| WireMask-RS | Ours | 6.13 ± 0.05 | 59.28 ± 1.48 | 60.60 ± 0.45 | 62.06 ± 0.22 | 2.19 ± 0.01 | 62.58 ± 2.07 | **8.20 ± 0.17** | 0/5/2 | 2.57 |
| WireMask-BO | Ours | 6.07 ± 0.14 | 59.17 ± 3.94 | 61.00 ± 2.08 | 63.86 ± 1.01 | 2.14 ± 0.03 | 67.48 ± 6.49 | 8.62 ± 0.18 | 0/3/4 | 2.86 |
| WireMask-EA | Ours | **5.91 ± 0.07** | **52.63 ± 2.23** | **57.75 ± 1.16** | 58.79 ± 1.02 | **2.12 ± 0.01** | **59.87 ± 3.40** | 8.28 ± 0.25 | | **1.43** |

denoted as WireMask-RS, WireMask-BO and WireMask-EA, respectively. For the employed method TuRBO [16] for BO, we use the common hyper-parameters. For the EA, its initial solution is set as the best among 100 random solutions. All experiments are run on two Intel Xeon Platinum 8171M CPUs, each with 26 cores and 52 threads.

**Main results.** Table 1 gives the detailed results of running each method using five random seeds. The runtime for each run of WireMask-BBO is set as 1,000 minutes. The results of analytical and RL-based methods are directly from [26]. Note that the analytical method NTUPlace3 is deterministic, and thus has no standard deviation. We compute the rank of each method on each chip as in [21], which are averaged in the last column of Table 1. WireMask-RS, WireMask-BO and WireMask-EA achieve the three highest ranks, disclosing the effectiveness of the proposed general framework WireMask-BBO. Among these three variants, WireMask-EA performs the best, which has the highest rank 1.43 and achieves the smallest HPWL value on 5 out of the 7 chips. RS relies solely on random sampling without leveraging any search history. BO performs well in many low-dimensional tasks (typically when the dimension $d \leq 20$ [17]), but suffers from the curse of dimensionality due to the time-consuming cost of updating the Gaussian process surrogate model and optimizing the acquisition function [8]. For example, in our experiments where $d$ (i.e., 2 times the number of macros) is always larger than 1000, EA can sample much more solutions (about 2–7 times as shown in Table 9 of Appendix B) than BO during the 1000-minutes running. Compared with any previous method, WireMask-EA is significantly better on at least 6 out of the 7 chips, by the Wilcoxon rank-sum test with significance level 0.05. It is outperformed by RePlace and DREAMPlace on the chip *adaptec4*, implying the suitability of analytical methods for this particular problem.

**Running time analysis.** As shown in Table 1, any variant of WireMask-BBO always performs better than the state-of-the-art MaskPlace [26], which ranks the highest among the compared previous methods. While [26] did not offer explicit training time details, the authors state a 200-minute convergence time in response to our inquiry. Next, we examine the efficiency of WireMask-BBO when compared with MaskPlace. Taking MaskPlace as the baseline, we plot the curve of HPWL over the wall clock time for WireMask-BBO in Figure 4. Each variant of WireMask-BBO surpasses MaskPlace very quickly. Particularly, WireMask-EA requires only 39, 8, 0.37, 0.62, 0.15, and 3 minutes, respectively, with an average of 8 minutes to surpass MaskPlace across six benchmark chips. This is notably faster than the 200-minute convergence time reported for MaskPlace. We have excluded the *bigblue4* chip in Figure 4, because the 1,000-minute search for WireMask-EA only completes its initialization on this large-scale chip. On the chips *adaptec3* and *adaptec4*, WireMask-EA actually outperforms MaskPlace after only a single greedy placement guided by wire mask, implying that MaskPlace fails to balance exploration and exploitation.

**Performance on congestion metric.** Aside from HPWL, congestion and density are also vital metrics as introduced in Section 2.1. As our framework WireMask-BBO maintains the non-overlapping property, density constraint does not need to be considered. Here, we evaluate placement congestion and compare it to the results of MaskPlace [26] and SP-SA [33] in Table 2. We normalize the congestion values by setting WireMask-EA's congestion to 1.00. The MaskPlace placements are provided by its authors[2], while the SP-SA and WireMask-BBO placement results are obtained from a

---

[2]All the MaskPlace evaluations are based upon the placement file provided by its authors. Note that the authors provided only one placement on each benchmark chip, and did not provide the results on *bigblue4*, which are marked by a '/' symbol in the following tables.

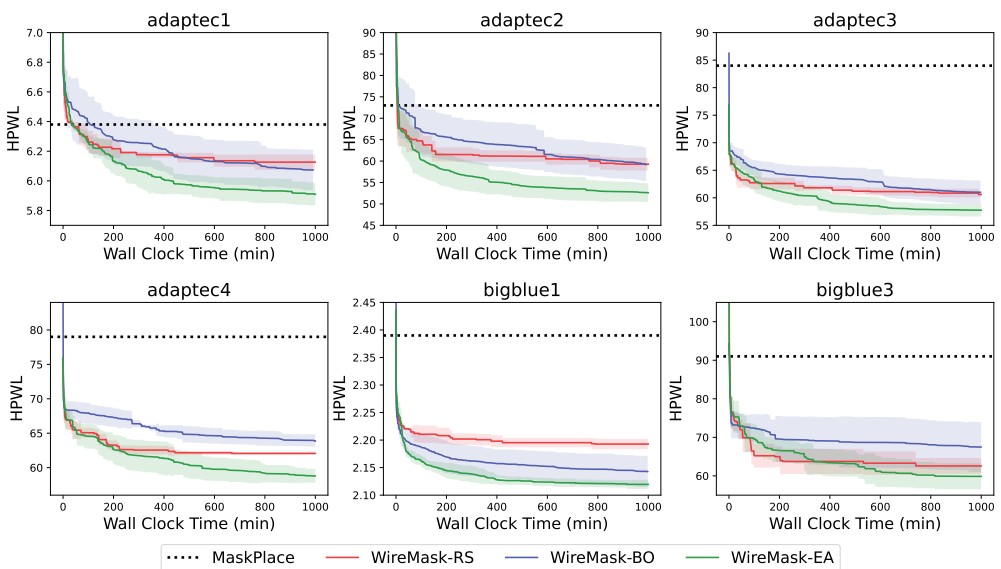

Figure 4: HPWL ($\times 10^5$) vs. wall clock time of WireMask-BBO, where the shaded region represents the standard error derived from 5 independent runs.

Table 2: Comparison on HPWL ($\times 10^5$) and Congestion (Cong.).

| Benchmark | adaptec1 | | adaptec2 | | adaptec3 | | adaptec4 | | bigblue1 | | bigblue3 | | bigblue4 | | Avg. Rank | |
| Metrics | HPWL | Cong. | HPWL | Cong. | HPWL | Cong. | HPWL | Cong. | HPWL | Cong. | HPWL | Cong. | HPWL | Cong. | HPWL | Cong. |
|---|---|---|---|---|---|---|---|---|---|---|---|---|---|---|---|---|
| SP-SA [33] | 18.77 | 4.21 | 117.63 | 1.34 | 122.30 | 2.03 | 102.54 | 1.54 | 5.30 | 2.09 | 136.16 | 1.50 | 19.00 | 2.72 | 4.86 | 4.71 |
| MaskPlace [26] | 6.56 | 1.22 | 79.98 | 1.63 | 79.32 | 1.25 | 75.75 | 1.17 | 2.42 | 1.63 | 82.61 | **0.86** | / | / | 4.00 | 3.67 |
| WireMask-RS | 6.09 | 1.01 | 57.31 | 1.04 | 60.91 | 1.04 | 60.02 | **0.98** | 2.18 | 1.09 | 63.86 | 1.01 | **8.21** | 1.10 | 2.29 | 2.43 |
| WireMask-BO | 6.14 | 1.02 | 55.31 | **0.99** | 58.67 | 1.06 | 61.67 | 1.06 | **2.10** | 0.94 | 68.88 | 1.19 | 8.39 | 1.07 | 2.29 | 2.43 |
| WireMask-EA | **5.81** | 1.00 | **49.32** | 1.00 | **56.56** | **1.00** | **58.79** | 1.00 | 2.12 | 1.00 | **60.88** | 1.00 | 8.45 | **1.00** | **1.43** | **1.57** |

1,000-minute search using the same random seed. The results show that though optimizing HPWL, WireMask-BBO can also achieve smaller congestion. The reason is that the RUDY approximation of congestion is sometimes positively related to the HPWL metric. Given a macro placement solution, the HPWL in Eq. (1) is computed as the sum of the rectangle's half-perimeter of hyper-edge, i.e., $\sum_{e_j \in E}(w_j + h_j)$, where $e_j$ denotes a hyper-edge, $E$ denotes the hyper-graph comprised of all hyper-edges, $w_j$ and $h_j$ denote the width and height of the rectangle corresponding to $e_j$, respectively. The RUDY measures the overall congestion on the canvas, and the congestion of each grid $g_i$ on the canvas is calculated by the cumulative impact of all hyper-edges encompassing the grid. Note that a hyper-edge $e_j$ will add an impact to each of its covered grids by $\frac{1}{w_j} + \frac{1}{h_j}$. Then, the overall congestion of all grids is $\sum_{g_i} \sum_{e_j \in E(g_i)} \frac{1}{w_j} + \frac{1}{h_j} = \sum_{e_j \in E} w_j \cdot h_j \cdot (\frac{1}{w_j} + \frac{1}{h_j}) = \sum_{e_j \in E}(w_j + h_j) = \text{HPWL}$, where $E(g_i)$ denotes the set of hyper-edges whose corresponding rectangle covers the grid $g_i$. The first equality holds because the number of times of a hyper-edge $e_j \in E$ enumerated in LHS is equal to the number of grids covered by it, which is $w_j \cdot h_j$. Thus, we can observe a positive relation between RUDY and HPWL. Besides our results, Table 4 in MaskPlace [26] and Table 4 in ChiPFormer [25] have also shown that the best HPWL can lead to the best congestion. However, a lower HPWL does not necessarily lead to a lower RUDY, because RUDY only considers the top-10% congested grids as introduced in Section 2.1.

**Full placement results.** Table 3 shows the results of different methods on the full placement task involving both macros and standard cells. We utilize PRNet [14] and DREAMPlace [28] as baselines, because both of them are designed to handle mixed-size placement tasks. The other five macro placement methods first optimize the macro placement, fix the macro positions, and then employ DREAMPlace to optimize the standard cell placement exclusively. The mean±std values are derived from DREAMPlace standard cell placement using five different random seeds. The results of PRNet are directly from [14]. Although WireMask-EA achieves the best performance on only one chip, i.e., *adaptec4*, it attains the highest average rank. Compared with the mixed-size placement optimization methods PRNet and DREAMPlace, WireMask-EA does not explicitly optimize the full

Table 3: Comparison of HPWL values ($\times 10^7$) on the full placement task involving both macros and standard cells. The best (smallest) mean value on each chip is bolded. The symbols '+', '$-$' and '$\approx$' indicate the number of chips where the result is significantly superior to, inferior to, and almost equivalent to WireMask-EA+DREAMPlace, respectively, according to the Wilcoxon rank-sum test with significance level 0.05.

| benchmark | adaptec1 | adaptec2 | adaptec3 | adaptec4 | bigblue1 | bigblue3 | bigblue4 | $+/-/\approx$ | Avg. Rank |
|---|---|---|---|---|---|---|---|---|---|
| PRNet [14] | **8.28** | 12.33 | 23.24 | 23.40 | 14.10 | 46.86 | **100.13** | 2/5/0 | 3.29 |
| DREAMPlace [28] | 11.10 ± 1.31 | 13.84 ± 1.74 | **17.03 ± 0.99** | 24.37 ± 1.13 | **10.06 ± 0.28** | **36.51 ± 0.56** | 175.86 ± 2.23 | 2/4/1 | 3.57 |
| SP-SA [33]+DREAMPlace | 10.18 ± 0.18 | 14.80 ± 0.01 | 30.63 ± 0.82 | 28.89 ± 0.02 | 10.70 ± 0.01 | 63.60 ± 0.12 | 203.79 ± 0.36 | 0/7/0 | 6.29 |
| MaskPlace [26]+DREAMPlace | 10.86 ± 0.01 | 12.98 ± 0.58 | 26.14 ± 0.07 | 23.52 ± 0.01 | 10.64 ± 0.01 | 54.98 ± 1.06 | / | 0/6/0 | 5.00 |
| WireMask-RS+DREAMPlace | 9.40 ± 0.02 | **9.07 ± 0.02** | 22.76 ± 0.01 | 22.09 ± 0.01 | 10.11 ± 0.03 | 42.58 ± 0.20 | 262.16 ± 0.08 | 2/4/1 | 3.00 |
| WireMask-BO+DREAMPlace | 9.19 ± 0.26 | 12.87 ± 0.01 | 26.61 ± 0.04 | 26.70 ± 0.01 | 10.56 ± 0.01 | 56.00 ± 3.32 | 122.28 ± 0.06 | 1/6/0 | 4.43 |
| WireMask-EA+DREAMPlace | 8.93 ± 0.01 | 9.20 ± 0.05 | 21.72 ± 0.01 | **20.51 ± 0.01** | 10.35 ± 0.02 | 42.52 ± 0.11 | 171.23 ± 0.48 | | **2.14** |

placement wirelength; thus, its best overall performance confirms the contribution of high-quality macro placement to an overall superior final placement.

**Fine-tuning results.** In fact, WireMask-BBO can be combined with any existing macro placement method for post-processing. The placement generated by any existing method can be treated as the initial solution of WireMask-BBO and then further improved. Table 4 shows the HPWL results of SP-SA and MaskPlace before and after 1,000 minutes of fine-tuning by WireMask-EA, with the average improvement ratio of 53.93% and 17.06%, respectively. Compared to Table 1, fine-tuning the placement of MaskPlace leads to the best HPWL value on the two chips *adaptec1* and *bigblue1*.

Table 4: HPWL ($\times 10^5$) values obtained after fine-tuning existing placements by running WireMask-EA for 1,000 minutes. The last column, Avg. Imp., denotes the average improvement ratio across six chips, obtained by comparing the HPWL values before and after fine-tuning.

| Method | adaptec1 | adaptec2 | adaptec3 | adaptec4 | bigblue1 | bigblue3 | Avg. Imp. |
|---|---|---|---|---|---|---|---|
| SP-SA [33] | 18.84 | 117.36 | 115.48 | 120.03 | 5.12 | 164.70 | |
| +WireMask-EA (1000min) | 6.02 ± 0.11 | 60.35 ± 4.41 | 57.88 ± 0.62 | 59.50 ± 0.92 | 2.21 ± 0.02 | 82.68 ± 18.17 | 53.93% |
| MaskPlace [26] | 6.56 | 79.98 | 79.32 | 75.75 | 2.42 | 82.61 | |
| +WireMask-EA (1000min) | 5.84 ± 0.10 | 61.43 ± 1.23 | 59.24 ± 2.71 | 60.35 ± 1.38 | 2.10 ± 0.01 | 74.93 ± 7.79 | 17.06% |

**Additional results.** We propose a post local search procedure to further improve final placement results. We investigate the influence of the two hyper-parameters of WireMask-BBO, i.e., number of partitions in chip canvas discretization, and adjustment order of macros in objective evaluation. For WireMask-EA, the best-performed variant of WireMask-BBO, the influence of different mutation operators is also examined. We compare the number of evaluations of WireMask-BBO methods in 1000 minutes. Furthermore, we test on the *ariane* RISC-V CPU design benchmark [48], still showing the clear superiority of WireMask-BBO over previous methods. We finally include comparisons with two concurrent advanced methods, i.e., ChiPFormer [25] and AutoDMP [4], and our WireMask-EA still maintains the superior performance. These results are provided in Appendix B.

## 5   Conclusion

This paper proposes the general framework WireMask-BBO for macro placement, which adopts a wire-mask-guided greedy genotype-phenotype mapping and can be equipped with any BBO algorithm. Extensive experimental results show that WireMask-BBO is clearly superior to previous packing-based, analytical, and RL-based methods. Furthermore, it can be combined with any existing macro placement method to further improve the final placement. Though showing significant potential, WireMask-BBO also has some limitations. First, it can only deal with macro placement, leaving standard cells for analytical placers. Second, its performance is limited for chips with a large number of macros due to the expensive objective evaluation. This, however, can be alleviated by equipping with high-dimensional BBO algorithms [8] or designing specific efficient BBO algorithms for macro placement, which is a very interesting future work. Our method does not have negative social impacts.

## Acknowledgement

We would like to thank Yao Lai from The University of Hong Kong for his generous help and valuable discussions, and the anonymous reviewers for their helpful comments. This work was supported by the National Key R&D Program of China (2022ZD0116600) and National Science Foundation of China (62022039). Chao Qian is the corresponding author.

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

# A    Experimental Settings

The detailed statistics of benchmarks are in Table 5. Each column in the table represents the following:

- #Macros: the number of macros.

- #Standard cells: the number of standard cells.

- #Hyper-edges: the number of hyper-edges.

- #Macro-related Hyper-edges: the number of hyper-edges that are related to Marco, which is used in the objective evaluation process of WireMask-BBO.

- Area Utilization Ratio: the proportion of the total area of the chip canvas that is occupied by the macros, calculated by dividing the area of the macros by the total area of the chip canvas. A higher Area Utilization Ratio to some extent indicates a harder placement problem, because we have to maintain the non-overlapping property.

- #Partitions: the number of partitions when discretizing the chip canvas into grids.

Table 5: Detailed statistics of different benchmarks.

| Benchmark | #Macros | #Standard cells | #Hyper-edges | #Macro-related Hyper-edges | Area Utilization Ratio | #Partitions |
|---|---|---|---|---|---|---|
| adaptec1 | 543 | 210,904 | 3,709 | 693 | 0.48 | 160 |
| adaptec2 | 566 | 254,457 | 266,009 | 4,201 | 0.63 | 158 |
| adaptec3 | 723 | 450,927 | 466,758 | 3,259 | 0.55 | 113 |
| adaptec4 | 1,329 | 494,716 | 515,951 | 2,949 | 0.48 | 108 |
| bigblue1 | 560 | 277,604 | 284,479 | 409 | 0.27 | 160 |
| bigblue2 | 23,084 | 534,782 | 577,235 | 33,223 | 0.38 | / |
| bigblue3 | 1,298 | 1,095,514 | 1,123,170 | 3,937 | 0.66 | 234 |
| bigblue4 | 8,170 | 2,196,183 | 2,229,886 | 22,223 | 0.37 | 273 |
| ariane | 932 | 0 | 12,404 | 12,404 | 0.78 | 357 |

All columns, except for #Partitions, are predetermined and unalterable in the benchmark.[3] The #Partitions, however, constitutes a crucial hyperparameter, which is determined heuristically based on the heights and widths of the macros and the canvas. To elaborate, in the case where multiple macros of identical size, e.g., $30 \times 20$, are present on a given chip, with the canvas size $9000 \times 9000$, the value of #Partitions would be set to $300$, thereby resulting in a grid size of $30 \times 30$ for each grid. The influence of #Partitions is discussed in Table 7 of Appendix B.2.

# B    Additional Results

## B.1    Post local search

As the genotype-phenotype mapping in the objective evaluation of WireMask-BBO is performed greedily, guided by wire mask, we propose a post local search procedure to modify the final generated macro placement, which will probably bring further improvement. For a final generated macro placement, the positions of all macros are sequentially adjusted; the position of each macro is adjusted to the best grid (ties are broken randomly) which reduces the current HPWL value most. Note that when adjusting the position of one macro, all the other macros are fixed, which is different from that in the objective evaluation of WireMask-BBO, which is only based on the positions of those macros that have been adjusted. We have applied this post local search procedure to the final placements generated by SP-SA [33], MaskPlace [26] and WireMask-BBO. The macros are adjusted sequentially according to the order employed by WireMask-BBO (i.e., lines 1–2 in Algorithm 1); this process is performed twice. Note that the adjustment order of macros can be arbitrary when applying this post local search process. The results in Table 6 show that the final placements can be further improved, which is expected because both MaskPlace and WireMask involve (probabilistically) greedy incremental placement guided by wire mask.

---

[3]Note that for the *bigblue2* chip, which has over 20,000 macros and 30,000 macro-related hyper-edges, a single objective evaluation by step-by-step placement costs more than an hour. Thus, we do not include this chip in our experiments as [26], and its #Partitions is denoted as /.

Table 6: HPWL values $(\times 10^5)$ obtained by post local search on existing placements. The last column, Avg. Imp., denotes the average improvement ratio across seven chips, obtained by comparing the HPWL values before and after post local search.

| benchmark | adaptec1 | adaptec2 | adaptec3 | adaptec4 | bigblue1 | bigblue3 | bigblue4 | Avg. Imp. |
|---|---|---|---|---|---|---|---|---|
| SP-SA [33] | 18.84 ± 4.62 | 117.36 ± 8.73 | 115.48 ± 7.56 | 120.03 ± 4.25 | 5.12 ± 1.43 | 164.70 ± 19.55 | 25.49 ± 2.73 | 25.62% |
| after_local_search | 12.52 ± 1.59 | 97.22 ± 8.34 | 104.46 ± 7.73 | 102.17 ± 5.50 | 2.70 ± 0.17 | 138.85 ± 14.53 | 14.94 ± 0.84 | |
| MaskPlace [26] | 6.56 | 79.98 | 79.33 | 75.75 | 2.42 | 82.61 | / | 9.69% |
| after_local_search | 6.15 ± 0.05 | 72.46 ± 3.69 | 74.18 ± 0.68 | 67.99 ± 1.13 | 2.22 ± 0.02 | 77.87 ± 1.54 | / | |
| WireMask-RS | 6.13 ± 0.05 | 59.28 ± 1.48 | 60.60 ± 0.45 | 62.06 ± 0.22 | 2.19 ± 0.01 | 62.58 ± 2.07 | 8.20 ± 0.17 | 3.57% |
| after_local_search | 5.99 ± 0.06 | 57.23 ± 1.75 | 59.22 ± 0.40 | 60.99 ± 0.82 | 2.17 ± 0.01 | 58.32 ± 3.57 | 7.64 ± 0.24 | |
| WireMask-BO | 6.07 ± 0.14 | 59.17 ± 3.94 | 61.00 ± 2.08 | 63.86 ± 1.01 | 2.14 ± 0.03 | 67.48 ± 6.49 | 8.62 ± 0.18 | 3.77% |
| after_local_search | 6.04 ± 0.12 | 54.24 ± 2.92 | 60.29 ± 2.29 | 62.96 ± 1.16 | 2.13 ± 0.01 | 63.87 ± 8.26 | 7.86 ± 0.21 | |
| WireMask-EA | 5.91 ± 0.07 | 52.63 ± 2.23 | 57.75 ± 1.16 | 58.79 ± 1.02 | 2.12 ± 0.01 | 59.87 ± 3.40 | 8.28 ± 0.25 | 3.32% |
| after_local_search | 5.89 ± 0.07 | 49.04 ± 2.35 | 57.41 ± 1.04 | 57.72 ± 0.76 | 2.11 ± 0.01 | 56.29 ± 5.23 | 7.66 ± 0.26 | |

## B.2 Hyper-parameter analysis

We empirically investigate the influence of the the hyper-parameters of WireMask-EA, i.e., the number of partitions when discretizing the chip canvas into grids, the order of adjusting the positions of macros in objective evaluation, and the mutation operator.

**Number of partitions.** In our main experiments, the number of partitions is heuristically determined based on detailed macro statistics and varies across different chips, ranging from 108 for *adaptec4* to 273 for *bigblue4*, as shown in Table 5. The first two rows of Table 7 show the results of WireMask-EA after increasing and decreasing the number of partitions by 100, respectively. The last row gives the results of the default WireMask-EA, achieving the highest average rank.

Table 7: HPWL values $(\times 10^5)$ obtained by WireMask-EA under different grid partition numbers. The symbol $\times$ denotes that the method fails to generate non-overlapping placements. The best (smallest) mean value on each chip is bolded. The symbols '+', '−', and '≈' indicate that the number of chips where the result is significantly superior to, inferior to, and almost equivalent to the default setting, respectively, according to the Wilcoxon rank-sum test with significance level 0.05.

| Method | adaptec1 | adaptec2 | adaptec3 | adaptec4 | bigblue1 | bigblue3 | bigblue4 | +/−/≈ | Avg. Rank |
|---|---|---|---|---|---|---|---|---|---|
| +100_grid_num | **5.88 ± 0.09** | 56.97 ± 2.16 | **57.74 ± 0.98** | 59.36 ± 1.10 | 2.30 ± 0.02 | 62.93 ± 3.36 | 8.32 ± 0.24 | 0/3/4 | 1.71 |
| −100_grid_num | 7.07 ± 0.18 | × | × | × | 3.06 ± 0.07 | 84.37 ± 4.14 | 10.14 ± 0.30 | 0/4/0 | 3.00 |
| default setting | 5.91 ± 0.07 | **52.63 ± 2.23** | 57.75 ± 1.16 | **58.79 ± 1.02** | **2.12 ± 0.01** | **59.87 ± 3.40** | **8.28 ± 0.25** | | **1.28** |

By increasing the grid partition number, more fine-grained genotype-phenotype mapping guided by wire mask can be achieved, but the computational overhead is also increased. If the grid partition number is too small, the mapping may even not lead to non-overlapping placements, as observed on the chips *adaptec2*, *adaptec3* and *adaptec4* after decreasing the grid partition number by 100. Thus, a good balance needs to be considered, when determining the grid partition number in practice.

**Order of adjusting the positions of macros.** In the objective evaluation of WireMask-BBO, the positions of macros are adjusted sequentially, in the decreasing order of the sum of area of each macro's connected cells, as shown in lines 1–2 of Algorithm 1. Here, we also test the random adjustment order and the order only considering the area of each macro itself, the results of which are shown in the first two rows, respectively, of Table 8.

Table 8: HPWL values $(\times 10^5)$ obtained by WireMask-EA using different orders to adjust the positions of macros in objective evaluation. The symbol $\times$ denotes that the method fails to generate non-overlapping placements. The best (smallest) mean value on each chip is bolded. The symbols '+', '−' and '≈' indicate that the number of chips where the result is significantly superior to, inferior to, and almost equivalent to default setting, respectively, according to the Wilcoxon rank-sum test with significance level 0.05.

| Benchmark | adaptec1 | adaptec2 | adaptec3 | adaptec4 | bigblue1 | bigblue3 | bigblue4 | +/−/≈ | Avg. Rank |
|---|---|---|---|---|---|---|---|---|---|
| random_order | 12.65 ± 0.68 | × | × | 93.25 ± 4.66 | 3.40 ± 0.09 | × | × | 0/3/0 | 3.00 |
| size_only_order | 6.32 ± 0.13 | **47.39 ± 1.02** | 61.18 ± 1.03 | **54.57 ± 1.69** | 2.16 ± 0.01 | 102.99 ± 7.69 | 10.80 ± 0.24 | 2/5/0 | 1.71 |
| default setting | **5.91 ± 0.07** | 52.63 ± 2.23 | **57.75 ± 1.16** | 58.79 ± 1.02 | **2.12 ± 0.01** | **59.87 ± 3.40** | **8.28 ± 0.25** | | **1.28** |

The random order always leads to the worst result, and may even fail to generate non-overlapping placements. Compared with using the order only considering each macro's area, the original WireMask-EA performs significantly better on five chips, and worse on the other two chips. Its best overall performance is expected, because the order considering the sum of area of each macro's connected cells is more closely related to the performance metric HPWL.

**Mutation operator.** For WireMask-EA, which performs the best among the three variants of WireMask-BBO, we also examine the influence of its mutation operator. We employed the swap mutation operator in our experiments, which randomly selects two macros and interchanges their coordinates. To further investigate, we implement the uniform mutation operator as well, which selects a macro at random and uniformly reallocates it on the chip canvas. Moreover, we combine these two mutation operators, with each being executed with a probability of 1/2. As shown in Figure 5, the utilization of the swap mutation operator alone (i.e., Swap_only) performs better than using the uniform mutation operator alone (i.e., Uniform_only) or their combination (i.e., Mix).

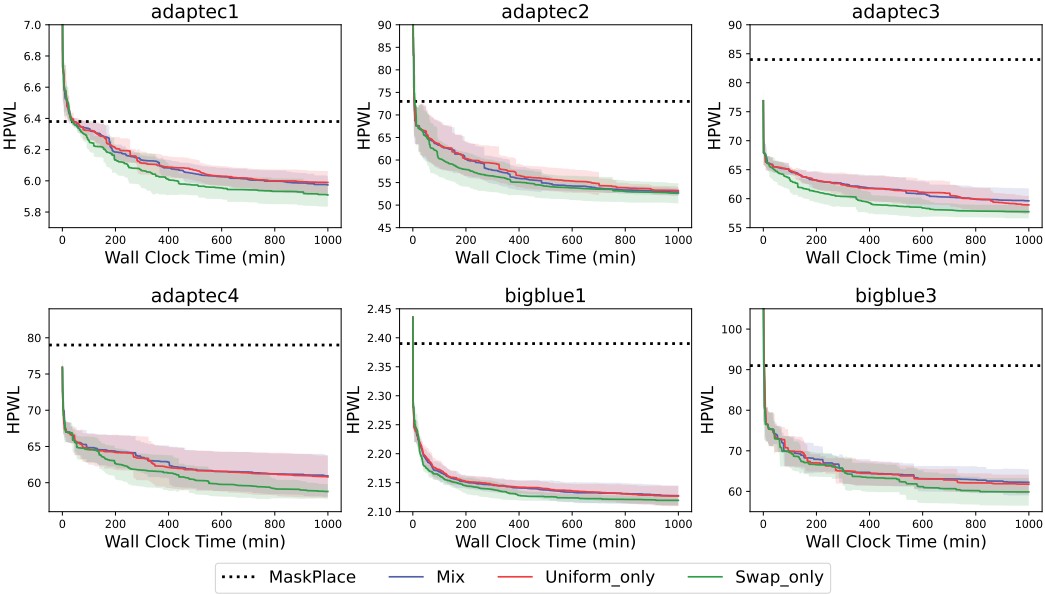

Figure 5: HPWL ($\times 10^5$) vs. wall clock time of WireMask-EA using different mutation operators, where the shaded region represents the standard error derived from 5 independent runs.

### B.3 Number of evaluations

Table 9 lists the number of evaluations used by WireMask-RS, WireMask-BO and WireMask-EA in Table 1, each of which runs for 1000 minutes. The number of evaluations of RS and EA is rather close, indicating that our EA operator is quite efficient. Besides, BO uses fewer evaluations than RS and EA, owing to the time-consuming cost of updating the Gaussian process model and optimizing the acquisition function.

Table 9: Number of evaluations of WireMask-RS, WireMask-BO and WireMask-EA in Table 1.

| 1000 min | adaptec1 | adaptec2 | adaptec3 | adaptec4 | bigblue1 | bigblue3 | bigblue4 |
|---|---|---|---|---|---|---|---|
| WireMask-RS | 3610 | 3689 | 5257 | 2791 | 7367 | 976 | 106 |
| WireMask-BO | 820 | 655 | 810 | 600 | 1580 | 540 | 59 |
| WireMask-EA | 3526 | 3540 | 5179 | 2685 | 7217 | 877 | 97 |

## B.4 Results on the *ariane* benchmark

We evaluate the performance of our WireMask-BBO framework on another benchmark, the *ariane* RISC-V CPU design [48], which has been previously studied in works such as [32, 26]. Our evaluation involves a comparison with three analytical methods and three RL methods, and the results are summarized in Table 10. The results of analytical and RL methods are directly from [26]. The analytical methods all fail to generate non-overlapping placement results on *ariane*, likely due to its high macro area utilization ratio of 0.78, which is the largest among all the benchmarks in Table 5. The overlapping issue is consistent with our discussion presented in Section 2.3. The RL-based methods show significantly worse performance compared to our proposed WireMask-BBO methods. The three different variants of WireMask-BBO are comparable based on the statistical test, with WireMask-EA demonstrating the lowest mean HPWL value.

Table 10: Comparison of HPWL values $(\times 10^5)$ on the *ariane* benchmark [48]. A '$\times$' symbol means that the method fails the legalization and thus cannot generate non-overlapping placement results. The best (smallest) mean value is bolded. The symbols '$+$', '$-$' and '$\approx$' indicate that the result is significantly superior to, inferior to, and almost equivalent to WireMask-EA, respectively, according to the Wilcoxon rank-sum test with significance level 0.05.

| Method | Type | ariane | $+/-/\approx$ |
|---|---|---|---|
| NTUPlace3 [12] | Analytical | $\times$ | |
| RePlace [13] | Analytical | $\times$ | |
| DREAMPlace [28] | Analytical | $\times$ | |
| Graph [32] | RL | $16.89 \pm 0.60$ | $-$ |
| DeepPR [15] | RL | $52.20 \pm 0.89$ | $-$ |
| MaskPlace [26] | RL | $14.63 \pm 0.20$ | $-$ |
| WireMask-RS | Ours | $9.90 \pm 0.18$ | $\approx$ |
| WireMask-BO | Ours | $9.71 \pm 0.46$ | $\approx$ |
| WireMask-EA | Ours | $\mathbf{9.68 \pm 0.61}$ | |

## B.5 Comparison with concurrent works

Recently, ChiPFormer [25] and AutoDMP [4] are proposed as new state-of-the-art methods in macro placement. For a comprehensive comparison, we also conduct experiments with these two concurrent works here.

**ChiPFormer.** ChiPFormer [25] incorporates an offline learning decision transformer to improve the generalizability. The pre-trained model provides placement results with acceptable quality within minutes, and converges to better results than MaskPlace [26] using fewer evaluations. We have compared our proposed WireMask-EA with ChiPFormer on ten chips, as outlined in Table 11. The experimental results show that our WireMask-EA clearly outperforms ChiPFormer on 9 circuits out of 10, no matter the number of evaluations is 1, 300 or 2k.

**AutoDMP.** AutoDMP [4] improves the efficient DREAMPlace [28], using Bayesian optimization to explore the configuration space and showing potential in real EDA application. We have included running time comparison between the state-of-the-art method AutoDMP [4] and our proposed WireMask-EA as shown in Figure 6. The results of MaskPlace (3k evaluations) and ChiPFormer (2k evaluations) are ploted by dotted lines in the figure. We can observe that on the two chip benchmarks, *adaptec1* and *bigblue1*, AutoDMP and WireMask-EA can surpass MaskPlace and ChiPFormer quickly, and WireMask-EA performs the best.

Table 11: HPWL values ($\times 10^5$) obtained by the recent proposed state-of-the-art method ChiP-Former [25] and our proposed WireMask-EA on ten chips, including *adaptec* and *bigblue* from ISPD2005 benchmark [35] and *ibm* from ICCAD2004 benchmark [3]. The number in () denotes the number of evaluations used. Each result consists of the mean and standard deviation of five runs. The best (smallest) mean value on each chip is bolded.

| Benchmark | ChiPFormer (1) | WireMask-EA (1) | ChiPFormer (0.3k) | WireMask-EA (0.3k) | ChiPFormer (2k) | WireMask-EA (2k) |
|---|---|---|---|---|---|---|
| adaptec1 | 8.87 ± 0.98 | **7.20 ± 0.34** | 7.02 ± 0.11 | **6.29 ± 0.07** | 6.62 ± 0.05 | **5.96 ± 0.08** |
| adaptec2 | 122.37 ± 22.61 | **111.04 ± 20.09** | 70.42 ± 2.67 | **61.25 ± 4.10** | 67.10 ± 5.46 | **53.88 ± 2.53** |
| adaptec3 | 107.11 ± 8.84 | **75.37 ± 2.93** | 78.32 ± 2.03 | **64.49 ± 1.69** | 76.70 ± 1.15 | **59.26 ± 1.30** |
| adaptec4 | 85.63 ± 7.52 | **75.63 ± 1.30** | 69.42 ± 0.54 | **64.52 ± 1.81** | 68.80 ± 1.59 | **59.52 ± 1.71** |
| bigblue1 | 3.11 ± 0.03 | **2.31 ± 0.06** | 2.96 ± 0.04 | **2.18 ± 0.01** | 2.95 ± 0.04 | **2.14 ± 0.01** |
| bigblue3 | 131.78 ± 17.36 | **99.20 ± 24.69** | 81.48 ± 4.83 | **64.51 ± 4.15** | 72.92 ± 2.56 | **56.65 ± 2.81** |
| ibm01 | 4.57 ± 0.27 | **3.76 ± 0.36** | 3.61 ± 0.08 | **2.92 ± 0.07** | 3.05 ± 0.11 | **2.39 ± 0.07** |
| ibm02 | 6.01 ± 0.41 | **5.13 ± 0.16** | 4.84 ± 0.17 | **3.86 ± 0.03** | 4.24 ± 0.25 | **3.56 ± 0.05** |
| ibm03 | **2.15 ± 0.17** | 3.10 ± 0.12 | **1.75 ± 0.07** | 2.20 ± 0.11 | **1.64 ± 0.06** | 1.69 ± 0.11 |
| ibm04 | 5.00 ± 0.14 | **3.60 ± 0.17** | 4.19 ± 0.11 | **2.93 ± 0.11** | 4.06 ± 0.13 | **2.62 ± 0.04** |

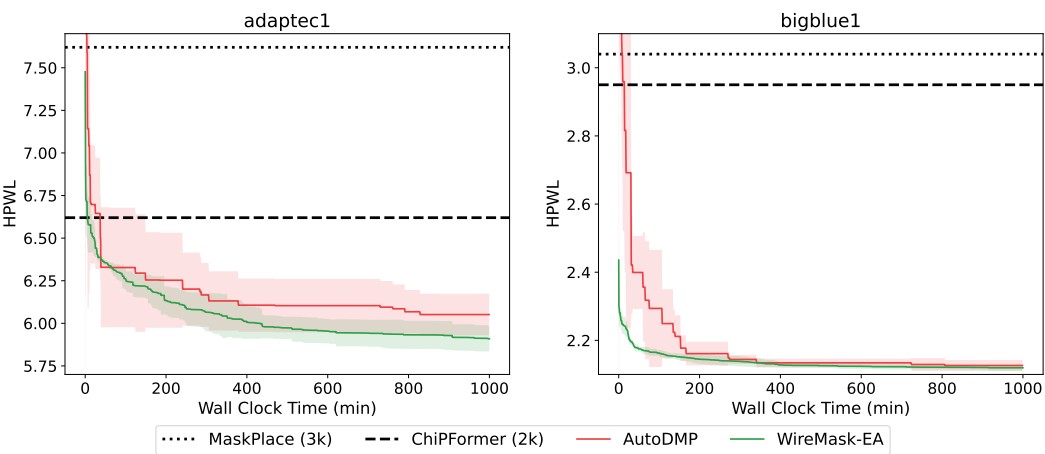

Figure 6: HPWL ($\times 10^5$) vs. wall clock time of WireMask-EA and AutoDMP [4], where the shaded region represents the standard error derived from 5 independent runs.

