# OpenReview forum: "Macro Placement by Wire-Mask-Guided Black-Box Optimization"
_NeurIPS.cc/2023/Conference — NeurIPS 2023 poster_

### Official Review · Reviewer_n9zj · 2023-07-05

**Soundness:** 2 fair
**Presentation:** 2 fair
**Contribution:** 2 fair
**Rating:** 5
**Confidence:** 4

**Summary:**

This paper proposes a new black-box optimization framework, called WireMask-BBO, for macro placement, which is an important problem in the electronic design automation (EDA) community. By using different black-box optimization algorithms, The experiments show it can achieve improvements (shorter half-perimeter wirelength (HPWL)) over previous methods.

**Strengths:**

The general framework WireMask-BBO proposed by this paper somewhat provides new angle for macro placement problem. The paper is well-written and it is easy to follow.

**Weaknesses:**

The scalability of the proposed method is doubtful because Bayesian optimization and evolutionary algorithm may not be suitable for large-scale problems.
The experiments are insufficient. The experiments don’t compare with the state-of-the-art macro placement method. The experiments don’t compare the runtime of the proposed method with other methods.

**Questions:**

(1)	How about the performance comparison with the state-of-the-art macro placement method in the experiments?
(2)	How about the runtime of the proposed method and compare it with other methods?
(3)	Is it necessary to tune hyperparameters extensively to achieve good results in the experiments?

**Limitations:**

Yes

---

> ### Author Rebuttal · Authors · 2023-08-10
>
> Thank you for your constructive comments and suggestions. Below please find our response.
>
> ### Q1 Runtime comparison; scalability of WireMask-BBO; BO and EA not suitable for large-scale problems?
>
> In our experiments, both the packing-based method SP-SA and our proposed WireMask-EA run for 1000 minutes. The results in Table 1 in the paper show that WireMask-EA achieves much better HPWL than SP-SA, implying that WireMask-EA is more efficient. The three analytical methods, NTUPlace3, RePlace and DREAMPlace, are very efficient, because the original complex problem is solved by relaxing to a mathematical programming problem. Compared with the state-of-the-art RL-based method MaskPlace, the results in Figure 4 in the paper have shown that WireMask-EA is much more efficient, which takes an average of 8 minutes to surpass the 200-minutes converged result of MaskPlace across six benchmark chips. Thanks to the suggestion of Reviewers 3gz2 and nUfA, we have compared the ChiPFormer method [1], recently published at ICML 2023. The results are shown in Table 1 of the accompanying PDF file. We can observe that after using the same number of evaluations, WireMask-EA outperforms ChiPFormer clearly on 9 out of 10 circuits. That is, WireMask-EA is more efficient than ChiPFormer. Thanks to your suggestion, we have implemented the AutoDMP method, recently published at ISPD 2023. The results in Figure 1 of the accompanying PDF file show that after using the same runtime, WireMask-EA achieves better HPWL than AutoDMP.
>
> Based on the above comparison, we can find that WireMask-EA has better scalability than previous methods except analytical ones. This can also be validated by the chip scales used in the experiments of these works. For example, only 3 data sets in ChiPFormer [1] contain more than 1000 macros (1329, 1293 and 1024, respectively); only two data sets in DeepPR+ [2] have over 1000 macros (1309 and 1227, respectively). When facing large data sets like bigblue4 with 8170 macros, these two works selected only hundreds of macros manually for placement. In the experiments of AutoDMP [3], only at most 320 macros are to be placed. Note that in our work, we have run the proposed framework WireMask-BBO on the full bigblue4 benchmark with 8170 macros, which is the largest scale reported so far, to the best of our knowledge.
>
> However, it must be acknowledged that our framework WireMask-BBO still struggle with very large-scale data sets, e.g., bigblue2 with 23084 macros. As WireMask-BBO can be equipped with any black-box optimization (BBO) algorithm, one direct way is to employ efficient Bayesian Optimization (BO) or Evolutionary Algorithm (EA) for high-dimensional scenario. Note that we only employed simple BBO techniques in the paper, which has led to superior performance over previous methods. Though BO and EA are traditionally inefficient for high-dimensional problems, many efforts have been put into this topic, generating many efficient BBO algorithms, e.g., RDHEBO [4] based on decomposition [4], ALEBO based on embbeding [5], MCTS-VS-BO based on variable selection [6], and self-guided evolutionary strategy [7]. We will revise to include the runtime comparison and add more discussion. Thank you.
>
>
> ### Q2 Comparison with AutoDMP [3].
>
> Thank you for pointing out this related method AutoDMP, which is recently published at ISPD'23. It is built mainly upon the efficient DREAMPlace, using BO to explore the configure space and showing potential in real EDA application. We have revised to add the comparison with AutoDMP. Due to time and computational resource limit, we were only able to run on the two chips, adaptec1 and biglue1. The results are shown in Figure 1 of the PDF file. We can observe that WireMask-EA outperforms AutoDMP. We will run AutoDMP on all the tasks, and include it in the final version. Thank you for your suggestion.
>
>
>
> ### Q3 Is it necessary to tune hyperparameters extensively to achieve good results in the experiments?
>
> We did not tune hyperparameters extensively. When applying the general framework WireMask-BBO, there are only two hyperparameters to be set, i.e., the number of partitions of the chip canvas and the employed BBO algorithm. In our experiments, the number of partitions is heuristically determined based on detailed macro statistics and varies across different chips, as shown in Table 4 of Appendix. Thanks to your suggestion, we have run the proposed WireMask-EA with 224 partitions for all the chips, which is consistent with the setting of MaskPlace. The results are shown in Table 3 of the PDF file. We can observe that WireMask-EA is still always better than MaskPlace and ChiPFormer, implying its robustness to the number of partitions of the chip canvas. Regarding the employed BBO algorithm, our experiments have shown that employing random search, a simple EA or the BO algorithm TurBO can all lead to superior performance over previous methods, implying the robustness of WireMask-BBO to the employed BBO algorithm. Note that for each employed BBO algorithm itself, we used its default hyperparameters for all chips. Thus, we believe our proposed framework WireMask-BBO is easy to use in practice. We hope our explanation can address your concerns. Thank you.
>
>
> References
>
> [1] ChiPFormer: Transferable Chip Placement via Offline Decision Transformer. ICML'23.
>
> [2] The Policy-Gradient Placement and Generative Routing Neural Networks for Chip Design. NeurIPS'22.
>
> [3] AutoDMP: Automated DREAMPlace-based Macro Placement. ISPD'23.
>
> [4] Are Random Decompositions all we need in High Dimensional Bayesian Optimisation? ICML'23.
>
> [5] Re-Examining Linear Embeddings for High-Dimensional Bayesian Optimization. NeurIPS'20.
>
> [6] Monte Carlo Tree Search based Variable Selection for High Dimensional Bayesian Optimization. NeurIPS'22.
>
> [7] Self-Guided Evolution Strategies with Historical Estimated Gradients. IJCAI'20.

---

> > ### Comment · Reviewer_n9zj · 2023-08-18
> >
> > Thanks for the clarification. VLSI placement is considered challenging and difficult due to the complex problem structure and the large scale (and will always be). So if a solution  cannot scale, then there must be other points that are extremely attractive to designers to make it acceptable, which is not significantly identified currently.

---

> > > ### Author Response · Authors · 2023-08-19
> > >
> > > Thanks for your feedback. However, we are confused about the concerns you expressed in the reply and are not sure if your concerns have been addressed. As to the weaknesses and questions you commented in the initial review, we believe that our response has addressed them. To be specific, we list your comments and a brief summary of our response below.
> > >
> > >  - Comparison with the recent method AutoDMP (published at ISPD'23). We have conducted additional experiments to compare our WireMask-EA with AutoDMP you mentioned. Besides, we have compared WireMask-EA with another SOTA method ChiPFormer (published at ICML'23) as suggested by Reviewer 3gz2 and Reviewer nUfA. Experimental results demonstrate the superior performance of our proposed WireMask-EA. Note that these two SOTA methods are considered concurrent works according to the NeurIPS rule and ChiPFormer is even not released when we submitted this work.
> > >
> > >  - Hyperparameter tuning. We introduced all the hyperparameters used in the paper. We did not tune hyperparameters extensively, and a common setting has been proven to show remarkable performance. We have added more detailed discussions on the settings of hyperparameters.
> > >
> > >  - Runtime analysis and scalability. We have provided detailed runtime analysis and comparison with related methods, including two SOTA methods AutoDMP and ChiPFormer, to reveal our runtime efficiency. For the scalability of our proposed WireMask-BBO, we have shown that WireMask-BBO performs better than other related works on large-scale problems, demonstrating the better scalability of our method.
> > >
> > > We fully agree with you that the scalability of the VLSI design method is indeed a significant challenge, which is, however, hardly to be fully addressed by a single work. Solving it step by step is more realistic from the perspective of scientific research. We believe our proposed WireMask-BBO has brought significant improvements to the existing SOTA methods, as shown in our experiments. Besides, WireMask-BBO provides a new perspective, i.e., solving VLSI by black-box optimization, which can provide more insight into the design of the VLSI method and can benefit from the progress of high-dimensional black-box optimization algorithms. Thus, we think our work has enough contributions to the community: the proposed WireMask-BBO not only achieves SOTA performance and can be used to fine-tune existing placements, but also has the potential to be a new viable direction for macro placement and promote further advances.
> > >
> > > **We hope that our response has addressed your concerns, but if we missed anything please let us know.**

---

> > > ### Author Response · Authors · 2023-08-21
> > > **Add further experiments to address your concerns.**
> > >
> > > Our proposed WireMask-BBO is a general framework for macro placement, which can be equipped with any black-box optimization (BBO) algorithm. Our experiments show that even employing simple BBO algorithms has led to superior performance over previous methods. The runtime analysis and comparison have shown that WireMask-BBO is more scalable than recent methods, which can also be validated by the largest chip scales used in the experiments of different works, e.g., 8170 marcos by WireMask-BBO vs. 1329 macros by ChiPFormer [1].
> > >
> > > As we claimed before, its versatility allows WireMask-BBO to benefit from the progress of high-dimensional BBO algorithms, e.g., efficient BBO algorithms for high-dimensional scenario can be employed to further improve the efficiency of WireMask-BBO. To show this, we select DropoutBO [2] arbitrarily, which is a Bayesian optimization (BO) algorithm for high-dimensional scenario based on random variable selection; and we have revised to implement WireMask-BBO equipped with DropoutBO on the benchmarks adaptec4 (with 1329 macros) and bigblue3 (with 1298 macros). The detailed results are shown in the following table, giving the HPWL value achieved every 50 search steps. We can clearly observe that compared with the BO algorithm TurBO [3] used in the paper, using DropoutBO can lead to significant improvement. Thus, we believe that the proposed general framework WireMask-BBO is scalable, and can bring a new viable direction for solving the important macro placement problem. We hope our further experiments and explanation can address your concerns. Thank you.
> > >
> > > | method           | dataset  | 50           | 100          | 150          | 200          | 250          | 300          | 350          | 400          | 450          | 500          |
> > > |------------|----------|--------------|--------------|--------------|--------------|--------------|--------------|--------------|--------------|--------------|--------------|
> > > | BO         | adaptec4 | 68.28 ± 1.31 | 67.76 ± 1.36 | 67.38 ± 1.4  | 66.38 ± 0.52 | 65.75 ± 0.46 | 65.23 ± 0.89 | 64.78 ± 1.18 | 64.47 ± 1.02 | 64.36 ± 1.00 | 64.25 ± 1.07 |
> > > | DropoutBO  | adaptec4 | 65.77 ± 1.65 | 65.3 ± 1.26  | 63.7 ± 0.45  | 63.25 ± 0.77 | 62.88 ± 1.03 | 61.5 ± 0.75  | 61.21 ± 0.91 | 61.18 ± 0.94 | 60.52 ± 0.97 | 60.08 ± 1.59 |
> > > | BO         | bigblue3 | 72.72 ± 3.27 | 71.9 ± 4.01  | 69.43 ± 5.97 | 69.31 ± 5.99 | 69.05 ± 6.17 | 68.7 ± 6.59  | 68.66 ± 6.59 | 68.61 ± 6.58 | 68.07 ± 6.35 | 67.83 ± 6.27 |
> > > | DropoutBO  | bigblue3  | 69.35 ± 3.07 | 64.41 ± 2.88 | 61.61 ± 2.31 | 61.17 ± 1.87 | 61.07 ± 1.92 | 60.7 ± 1.4   | 60.15 ± 1.81 | 59.58 ± 1.04 | 59.53 ± 0.96 | 59.03 ± 1.03 |
> > >
> > > References
> > >
> > > [1] ChiPFormer: Transferable chip placement via offline decision transformer. ICML'23.
> > >
> > > [2] High dimensional Bayesian optimization using dropout. IJCAI'17.
> > >
> > > [3] Scalable global optimization via local Bayesian optimization. NeurIPS'19.

---

### Official Review · Reviewer_KNNs · 2023-07-06

**Soundness:** 2 fair
**Presentation:** 3 good
**Contribution:** 2 fair
**Rating:** 5
**Confidence:** 4

**Summary:**

This paper presents a framework using BBO for macro placement in VLSI designs. Any placement solution for a set of macros can be optimized using the wire masks (presented in a prior work using RL) where the optimization goal is to minimize the HPWL of the output. In addition to random inputs, the framework can also be used for further enhancement of any existing solutions.

**Strengths:**

The work tackles a critical problem in VLSI designs.

The idea of casting the placement problem to a BBO is interesting and novel.

This framework can be used to further improve any existing solutions for the macros. This can be used as another step in PnR with reasonable runtime.

Overall, the paper explains the problem, the existing solutions, and the proposed work clearly.

**Weaknesses:**

The idea of casting the placement problem to a BBO is interesting and novel. However, as the authors admit in Section 1 that this work does not develop any new BBO algorithm. To the reviewer, the experimental results are not extensive and convincing; this is described in the limitation section.

The paper also states in the conclusion that it can only place macros but not standard cells without explanation, which limits the application of the proposed work to the actual VLSI problem. On the other hand, this framework can be used to further improve any existing solutions for the macros. This can be used as another step in PnR with reasonable runtime.

**Questions:**

Why is EA-based framework better than the other two BBO algorithms?

Why cannot the proposed framework used for standard cell placement? Can't you use a more fine-grained canvas for the problem, where the standard cells are larger than the grid size?

**Limitations:**

The computational time for the proposed work seems to be high. Therefore, in the experimental section, the paper does not include the complex benchmark that has thousands of macros to be placed. This contradicts with the claim the paper that this approach is scalable.

The evaluation of the proposed method only uses 5 random seeds. It would be more convincing if the paper includes more experimental data.

The proposed framework cannot place the standard cells in the design. Therefore, the comparison between this work and the existing methods seems to be unfair.

---

> ### Author Rebuttal · Authors · 2023-08-10
>
> Thank you for your valuable comments. Below please find our response.
>
> ### Q1 Does not develop any new BBO algorithm.
>
> We want to emphasize that our main contribution is introducing the general framework WireMask-BBO for macro placement, while not developing new BBO algorithms. WireMask-BBO can be equipped with any BBO algorithm. Our experiments show that even employing simple BBO techniques has led to superior performance over previous methods, suggesting that WireMask-BBO may be a new viable direction for macro placement. We agree that developing new efficient BBO algorithms under the framework WireMask-BBO is important for handling very large-scale circuits, as stated in our limitation part. But we believe that the contribution of proposing WireMask-BBO is significant enough. In fact, we plan to open-source WireMask-BBO and use it as an optimization benchmark to encourage the invention of more efficient BBO algorithms for solving macro placement problems, as well as broaden the application scenario of BBO.
>
> ### Q2 Why cannot be used for standard cell placement?
>
> As you indicated, the proposed framework can be naturally applied to standard cell placement by using a fine-grained canvas where the number of grids is larger than that of standard cells. However, the number of standard cells can be million, resulting in a very large search space. Furthermore, the wire-mask-guided greedy procedure for objective evaluation will be very expensive, which requires the calculation of wire mask for each cell. A more fine-grained canvas will also increase the wire mask computation time. Thus, our proposed WireMask-BBO currently cannot deal with standard cells. In fact, the limitation is shared with those methods based on step-by-step placement formulation, including Graph Placement, DeepPR, MaskPlace and ChiPFormer. We will revise to add some discussion. Thank you.
>
> ### Q3 Why is EA better than the other two BBO algorithms?
>
> RS relies solely on random sampling without leveraging any search history. BO performs well in many low-dimensional tasks (typically when the dimension $d \leq 20$ [1]), but suffers from the curse of dimensionality due to the time-consuming cost of updating the Gaussian process surrogate model and optimizing the acquisition function [2]. For example, in our experiments where $d$ (i.e., 2 times the number of macros) is always larger than 1000, EA can sample much more solutions (about 2--7 times) than BO during the 1000-minutes running. For the EA, we designed a specific mutation operator, which may also contribute its efficiency. We will revise to add more explanation. Thank you.
>
> ### Q4 Computational time high? Scalability?
>
> Please refer to Q1 in the response to Reviewer n9zj due to space limitation.
>
> ### Q5 More random seeds?
>
> Five random seeds are commonly used in previous works, e.g., MaskPlace and the recently proposed ChiPFormer [3]. However, we agree that using more random seeds would be better. Thanks to your suggestion, we have conducted additional experiments with 30 random seeds, testing WireMask-EA using 2000 evaluations on ibm01 and ibm02 benchmark. The obtained HPWL values (mean ± std.) are 2.48 ± 0.13 and 3.60 ± 0.11, respectively. When using 5 random seeds, they are 2.39 ± 0.07 and 3.56 ± 0.05, respectively. We believe that the slight difference will not affect the conclusions. For example, the results of the state-of-the-art method ChipFormer using 5 random seeds are 3.05 ± 0.01 and 4.24 ± 0.25, respectively, as shown in Table 1 of the PDF file; we can observe that they have clear gaps with the results of WireMask-EA. Due to time and computational resource limit, we were only able to run more random seeds on these two tasks. We will cover a wider range of tasks in the revised version. Thank you for your suggestion.
>
> ### Q6 The proposed framework cannot place the standard cells in the design. Therefore, the comparison between this work and the existing methods seems to be unfair.
>
> The previous methods SP-SA, Graph Placement, DeepPR+, MaskPlace, ChiPFormer, and the proposed WireMask-BBO all concentrate on macro placement. When comparing the results of full placement, the same DREAMPlace is applied to the macro placement generated by each method for the subsequent standard cell placement. Thus, the comparison is fair. In fact, such a flow has also been adopted in previous works, e.g., MaskPlace and ChiPFormer. We will revise to make it clearer.
>
>
> References
>
> [1] A Tutorial on Bayesian Optimization. 2018.
>
> [2] A Survey on High-dimensional Gaussian Process Modeling with Application to Bayesian Optimization. ACM TELO'22.
>
> [3] ChiPFormer: Transferable Chip Placement via Offline Decision Transformer. ICML'23.

---

> > ### Comment · Reviewer_KNNs · 2023-08-21
> >
> > The authors answered my questions. The proposed method opens a new research direction in VLSI placement with promising experimental results. However, this method cannot scale to the existing real-world VLSI designs, and the authors admitted that it will be hard to scale to the existing VLSI problems. It would be good to include a description in the paper of how to include this method in the existing VLSI design flows. I have raised my rating to 5.

---

> > > ### Author Response · Authors · 2023-08-21
> > > **Thanks for your reply.**
> > >
> > > Thanks for your reply! One direct way is to apply the proposed method for macro placement, and then use DREAMPlace for standard cell placement, which has been adopted in many works, e.g., DeepPR, MaskPlace and ChiPFormer. We will include a description in the final version. Thank you.

---

### Official Review · Reviewer_nUfA · 2023-07-10

**Soundness:** 3 good
**Presentation:** 3 good
**Contribution:** 3 good
**Rating:** 7
**Confidence:** 5

**Summary:**

The authors propose a new placement method that is based on the black-box framework. The framework leverages the wire mask-guided information and can achieve significant placement results compared with the state-of-the-art methods.

**Strengths:**

1. The novel method is based on black-box optimization, which has not been implemented on the placement task before. Although many RL methods have been proposed, they are not very efficient enough. Black-box optimization might be a viable direction.

2. The wire mask as the guide for generating phenotype representation is also very novel. It can quickly render a suitable solution based on the initial representation, improving efficiency.

3. The experiments are very comprehensive and solid, showing that the performance of the proposed method can consistently outperform existing methods.

4. The paper writing is well-written and easy to understand.

5. The code is open-source, improving reproducibility.


**Weaknesses:**

1. The full placement cannot surpass the DREAMPlace, which means the proposed method can only work well in macro placement.

2. The reasons for the improvement in the congestion metric are not clear. The method does not consider the congestion metric in its search process.

3. The recent work [1] based on Maskplace should also be discussed in the related work part.

[1] Lai Y, Liu J, Tang Z, et al. Chipformer: Transferable chip placement via offline decision transformer.


**Questions:**

1. Have you tested the effect of the number of initial states on the final results?

2. Why your method can get better congestion results when you do not consider it in your method.

3. The efficient comparison only provides the clock time. However, the number of search steps is also substantial. Could you provide how many steps you use for each circuit (or how much clock time is consumed to perform a step)?


**Limitations:**

Yes.

---

> ### Author Rebuttal · Authors · 2023-08-10
>
> Thank you for your valuable and positive comments. Below please find our response.
>
> ### Q1 The full placement cannot surpass DREAMPlace.
>
> In Table 6 of Appendix B.2, though DREAMPlace achieves the best HPWL in 3 out of 7 tested chips, our proposed WireMask-EA has significant improvements in four chips, significant disadvantages in two chips, and no significant difference in one chip, according to the Wilcoxon rank-sum test with significance level 0.05. This indicates an overall superiority of our proposed method. However, we cannot guarantee consistent superiority over DREAMPlace in all data sets. This is because for the full placement task, WireMask-EA first conducts macro placement and then uses DREAMPlace for standard cell placement, while DREAMPlace considers the placement of macros and standard cells simultaneously, which may lead to favorable results. One of our future work is to consider standard cell placement in our framework, as stated in the limitation part. Thank you.
>
>
> ### Q2 Why congestion better?
>
> Please refer to Q2 in general response.
>
> ### Q3 Discussion on ChiPFormer [1].
>
> Please refer to Q1 in general response.
>
> ### Q4 Influence of the number of initial states?
>
> The Evolutionary Algorithm (EA) adopted in our paper starts from one single initial solution, and iteratively improves it by mutation and selection. The initial solution is generated by selecting the best from a pool of 100 random solutions. We guess that you may wonder the influence of the pool size here. Thanks to your suggestion, we have tested WireMask-EA on the chip adaptec1 using the pool size of 1, 10, 20, 50, 200, and 2000. For each run of WireMask-EA, the total number of evaluations used is set to 2000, which has included the number of evaluations (i.e., the pool size) for initialization. Note that WireMask-EA with the pool size of 2000 is just WireMask-RS which performs random search. The final HPWL values are shown below. We can observe that as the pool size increases gradually, which will lead to a better initial solution, WireMask-EA achieves better (smaller) HPWL. However, when the pool size is large enough (e.g., 200 here), the performance of WireMask-EA starts to degrade, which is expected because too many evaluations are used for initialization (which performs random search) and the following exploration by EA tends to be insufficient. We used the pool size 100 in our experiments. Meanwhile, we can observe that WireMask-EA with any pool size can surpass the state-of-the-art method ChiPFormer, which achieves the HPWL value 6.62 ± 0.05 after using 2000 evaluations, as shown in the first line of Table 1 of the accompanying PDF file.
>
> | Pool size for initialization | 1           | 10          | 20          | 50          | 100         | 200         | 2000        |
> | ---                          | ---         | ---         | ---         | ---         | ---         | ---         | ---         |
> | adaptec1                     | 6.50 ± 0.34 | 6.22 ± 0.10 | 6.14 ± 0.06 | 6.11 ± 0.09 | 5.96 ± 0.08 | 5.98 ± 0.06 | 6.13 ± 0.05 |
>
> ### Q5 How many steps used for each circuit?
>
> Thanks for indicating this issue. In the experiments, our method is run for 1000 minutes on each chip. The number of search steps used for each chip is shown below, which will be included in the revised version. Thanks to your suggestion in Q3, we have compared WireMask-EA with ChipFormer under the same number of search steps, as shown in Table 1 of PDF file. We can observe that using 1, 300 or 2k steps, WireMask-EA outperforms ChipFormer in 9 out of 10 chips. We will revise to add some discussion. Thank you.
>
> | 1000 min    | adaptec1 | adaptec2 | adaptec3 | adaptec4 | bigblue1 | bigblue3 | bigblue4 |
> | ----------- | -------- | -------- | -------- | -------- | -------- | -------- | -------- |
> | WireMask-RS | 3610     | 3689     | 5257     | 2791     | 7367     | 976      | 106      |
> | WireMask-BO | 820      | 655      | 810      | 600      | 1580     | 540      | 59       |
> | WireMask-EA | 3526     | 3540     | 5179     | 2685     | 7217     | 877      | 97       |
>
>
> References
>
> [1] ChiPFormer: Transferable Chip Placement via Offline Decision Transformer. ICML'23.

---

> ### Comment · Reviewer_nUfA · 2023-08-20
>
> Thank you for your reply. The rebuttal addresses my concerns. I think this work is novel and experimental sufficiency. Although there is reviewer concern about scalability, from the research perspective, this work still has great potential with the black box optimization method. So, I still recommend this paper for acceptance.

---

> > ### Author Response · Authors · 2023-08-20
> > **Thanks for your appreciation.**
> >
> > Thanks for your appreciation. We are glad to hear that your concerns have been addressed. We will make sure to include the added results and discussion in the final version. Thank you.

---

### Official Review · Reviewer_3gz2 · 2023-07-12

**Soundness:** 3 good
**Presentation:** 3 good
**Contribution:** 3 good
**Rating:** 6
**Confidence:** 4

**Summary:**

This paper proposes a novel black-box optimization (BBO) framework, namely WireMask-BBO, for macro placement in chip design. Specifically, it devises a post-processing technique that legalizes any searched placement solution while optimizing the half-perimeter wirelength (HPWL). The post-processing technique allows us to perform BBO algorithms to search for solutions with better HPWL. Experiments demonstrate that WireMask-BBO outperforms previous state-of-the-art (SOTA) methods, achieving better HPWL performance in less time.

**Strengths:**

1.	This paper explores BBO methods for macro placement, which may provide a new insight for the research community.
2.	The proposed post-processing technique is simple yet effective. It also has a good versatility because it can be combined with other placement methods and BBO methods.

**Weaknesses:**

1.	The propose of the post-processing is purely heuristic. The motivation is unclear and intuitive explanation for the advances is insufficient.
2.	The proposed method only targets on optimizing HPWL, without explicitly considering other important metrics like routing wirelength or congestion. It does not consider cells and routing either. Moreover, the framework can be hardly transferred to tasks with those metrics under consideration, which limits its real application in EDA.
3.	Because introducing BBO-based methods is one of the core contributions of this paper, the authors may want to illustrate the implementation of BBO algorithms for macro placement more detailly.
4.	In Algorithm 1, the macros are ordered decreasingly according to areas, while in Figure 3, the smaller macro-A is considered first.
5.	A recent related work [1] is missing.
[1] Lai Y, Liu J, Tang Z, et al. Chipformer: Transferable chip placement via offline decision transformer. ICML 2023.

**Questions:**

1.	Since only HPWL is considered as the objective, why does WireMask-BBO also outperforms MaskPlace in congestion?
2.	What are the results of routing wirelengths?
3.	Under this framework, how to take cell placement or some important yet computationally insufficient metrics into consideration?

---

> ### Author Rebuttal · Authors · 2023-08-10
>
> Thank you for your constructive comments. Below please find our response.
>
> ### Q1 The proposed method is heuristic; insufficient intuitive explanations.
>
> We want to clarify our motivation of designing the proposed framework. To efficiently improve the HPWL of a solution and guarantee non-overlapping, we first design a greedy procedure guided by wire mask, which sequentially adjusts each macro of the solution to a position with the minimum marginal increment on HPWL. The adjustment order of each macro is determined by the area of all the cells connected with it, and all macros will be adjusted sequentially in the decreasing order of their corresponding computed areas, which is because a macro with a larger computed area implies connecting with more large cells and thus is intuitively more important. In the adjustment of each macro, if the position with the minimum marginal increment on HPWL is not unique, the one closest to its original position is selected, which can utilize some global information of the original placement to reduce the risk of getting trapped in local optima due to the greedy nature. Based on this greedy procedure, we further apply BBO algorithms for exploration, to search for solutions with better HPWL. We believe our proposed framework is intuitive and reasonable. We will add more explanations in the paper. Thank you.
>
> Q2 Optimize HPWL of macros only; how to take cell placement or more metrics into consideration?
>
> Thanks for your question, which is indeed a challenge faced by the area of macro placement. Following previous works (Graph Placement, DeepPR and ChiPFormer), we selected HPWL as the objective to be optimized, because it is a good approximation for wirelength and is also efficient to compute. Meanwhile, more metrics can be considered in our proposed BBO framework by directly treating them as the optimization objective. But there are two issues to be tackled before application. Firstly, as you indicated, the metric may involve cell placement and can be computationally expensive. To address this issue, one possible way is to employ advanced BBO techniques such as SAASBO [1] and Guided-ES [2], which can efficiently optimize expensive functions. Note that our framework is general and can be equipped with any BBO algorithm. For example, we only employed simple BBO techniques in the paper, which, however, has led to superior performance over previous methods. Secondly, our greedy procedure requires a criterion (e.g., HPWL used in the paper) which can guide the improvement of a solution and can also be computed step by step. For this issue, we may design a surrogate metric to approximate the true one, e.g., we can use RUDY to approximate the congestion. We will revise to add some discussion, and treat it as an important future work to explore. Thank you.
>
> ### Q3 More details of the implementation of BBO algorithms.
>
> WireMask-BBO employs a wire-mask-guided greedy adjustment procedure (generating feasible solutions with high quality efficiently), to serve as a black-box function, and adopts RS, BO and EA for optimization. To be specific, RS generates solutions by allocating all macros' positions randomly, and records the historical best. BO establishes a surrogate model, and samples a new solution by optimizing an acquisition function based on the surrogate model. We used the specific BO, TuRBO, in the paper. EA maintains a population of solutions, and iteratively improves the population by recombination and mutation. In the paper, we used a simple EA which maintains only one solution and uses mutation only. We designed the mutation operator by randomly selecting two macros and exchanging their coordinates in a solution. We are sorry for not illustrating them detailly enough, and will add more discussions in the paper. Thank you for your suggestion.
>
> ### Q4 Macro-A considered first in Figure 3?
>
> As described in line 188, Page 5, the placement order of a macro is determined by the sum of area of all its connected macros (including itself), instead of the macro's own area. In Figure 3, the macro A is connected to both B and C, which has the largest connected area, and thus is adjusted first. We will revise to make it clearer.
>
> ### Q5 Comparison with ChiPFormer.
>
> Please refer to Q1 in general response.
>
> ### Q6 Why congestion better?
>
> Please refer to Q2 in general response.
>
> ### Q7 Routing wirelengths?
>
> Yes, routing wirelength is a critical metric, whose evaluation is, however, expensive because it must be evaluated after standard cell placement and routing, and the evaluation itself is time-consuming. This is the motivation that we use HPWL as a surrogate, as also did in many previous works. Thanks to your suggestion, we have tried to report the routing wirelengths.
>
> For the routing task, our currently adopted benchmarks (i.e. ISPD05 and ICCAD04) are not supported by open source router OpenROAD or NCTU-GR 2.0. Thus, we turn to the DAC2012 benchmark [3] for routing wirelength evaluation with DREAMPlace for standard cell placement and NCTU-GR 2.0 for routing. For the proposed algorithm WireMask-EA, we first use it to generate a macro placement, and then adopt DREAMPlace for the subsequent standard cell placement. For DREAMPlace, we set all macros to be movable, and optimize macros and standard cells together. The full placement HPWL and the total wirelength results reported by NCTU-GR 2.0 are shown in Table 2 in the PDF. Due to time limitation, we select superblue19 with 286 macros for test. The results show that WireMask-EA achieves both better HPWL and routing wirelength. We will run the experiments on more benchmark chips from DAC2012, and add them into the final version. Thank you.
>
> References
>
> [1] High-Dimensional Bayesian Optimization with Sparse Axis-Aligned Subspaces. UAI'21.
>
> [2] Guided Evolutionary Strategies: Augmenting Random Search with Surrogate Gradients. ICML'19.
>
> [3] The DAC 2012 Routability-Driven Placement Contest and Benchmark Suite. DAC'12.

---

> > ### Comment · Reviewer_3gz2 · 2023-08-14
> > **Thank you. Increase my score to 6.**
> >
> > I appreciate the authors' efforts in responding to my questions. I hope the authors will revise the paper accordingly. Though there are still some limitaions as we have discussed, I think the proposed framework will benefit the research community, and it deserves further study.
> >
> > I have raised my score from 5 to 6.

---

> > > ### Author Response · Authors · 2023-08-15
> > > **Thank you! We are working on revising the paper.**
> > >
> > > Thanks for your reply! We will make sure to revise the paper according to all reviewers’ comments and suggestions, and incorporate the added results in our revision.

---

### Author Rebuttal · Authors · 2023-08-10

## General response

We are very grateful to the reviewers for carefully reviewing our paper and providing constructive comments and suggestions. We have revised the paper carefully according to the comments and suggestions, but we cannot upload the paper due to the NeurIPS' rule this year. Our response to individual reviewers can be found in the personal replies, but we also would like to make a brief summary of revisions about experiments for your convenience.

- We add the comparison with two important state-of-the-art methods, ChiPFormer [1] and AutoDMP [2].

- We add more benchmarks to test different methods, i.e., ibm series from ICCAD'04 benchmark [3].

- We add some results of routing wirelength.

- We add the analysis of the initial pool size and the partition number of the canvas for WireMask-EA.

- We add more random seeds (from 5 to 30) on some benchmarks.

- We add the details of the number of search steps during the 1000-minutes optimization of the proposed WireMask-BBO.

Below are responses to some common questions.

### Q1 Why not consider ChiPFormer [1]?

ChiPFormer, a recent paper published at ICML 2023 and initially released on arXiv on June 26, 2023, has gained our attention when it was published. However, due to its release date falling after the NeurIPS 2023 submission deadline, we were unable to include it in our manuscript.

ChiPFormer adopts an offline RL method, focusing on the HPWL metric of macro placement during optimization, and is equipped with a mixed-size placement workflow. The method demonstrates remarkable performance on various chip placement tasks. We acknowledge that incorporating a comparison with ChiPFormer would enhance the comprehensiveness of our work.

To ensure a fair evaluation, we promptly reached out to the authors of ChiPFormer upon reading their paper and obtained a standardized processed dataset (named ibm01--ibm04). We compared our proposed WireMask-EA with ChiPFormer on ten chips, as outlined in Table 1 of the accompanying PDF file. Notably, WireMask-EA outperforms ChiPFormer clearly on 9 out of 10 circuits, regardless of the number of evaluations employed (1, 300, or 2k).

We will discuss ChiPFormer in our paper, as well as include experimental comparisons. Thank you for your valuable feedback.

### Q2 Why congestion is better when only HPWL is optimized?

We feel sorry that we did not provide a persuasive explanation for such an important scenario in our paper, but we try to explain it here. We choose the widely adopted RUDY (Rectangular Uniform wire DensitY) to approximate congestion. Our conclusion is that the RUDY approximation of congestion is sometimes positively related to the HPWL metric, by analyzing the computation of HPWL and RUDY. Given a macro placement solution, the HPWL is computed as the sum of the rectangle's half-perimeter of each net (hyper-edge), i.e., $\sum_{e_j \in E} (w_j+h_j)$, where $e_j$ denotes a net, $E$ denotes the hyper-graph comprised of all nets, $w_j$ and $h_j$ denote the width and height of the rectangle corresponding to $e_j$, respectively. The RUDY measures the overall congestion on the canvas, and the congestion of each grid $g_i$ on the canvas is calculated by the cumulative impact of all nets encompassing the grid. Note that a net $e_j$ will add an impact to each of its covered grids by $\frac{1}{w_j} + \frac{1}{h_j}$. Then, the overall congestion of all grids is $\sum_{g_i} \sum_{e_j \in E(g_i)} \frac{1}{w_j} + \frac{1}{h_j}=\sum_{e_j \in E} w_j\cdot h_j\cdot (\frac{1}{w_j} + \frac{1}{h_j})=\sum_{e_j \in E} (w_j+h_j)=\mathrm{HPWL}$, where $E(g_i)$ denotes the set of nets whose corresponding rectangle covers the grid $g_i$. Note that the first equality holds because the number of times of a net $e_j \in E$ enumerated in LHS is equal to the number of grids covered by it, which is $w_j \cdot h_j$. Thus, we can observe a positive relation between RUDY and HPWL. Besides our empirical results, Table 4 in MaskPlace [4] and Table 4 in ChiPFormer [1] have also shown that the best HPWL can lead to the best congestion. However, we should also note that a lower HPWL does not necessarily lead to a lower RUDY, because RUDY only considers top-10\% congested grids. We will revise to add more discussion. Thank you.


References


[1] ChiPFormer: Transferable Chip Placement via Offline Decision Transformer. ICML'23.

[2] AutoDMP: Automated DREAMPlace-based Macro Placement. ISPD'23.

[3] ICCAD’04 Mixed-size Placement Benchmarks. 2009.

[4] MaskPlace: Fast Chip Placement via Reinforced Visual Representation Learning. NeurIPS'22.

---

### Decision · Program_Chairs · 2023-09-21

**Decision:**

Accept (poster)

**Comment:**

This paper proposes a general learning framework based on black-box optimization (BBO) methods for macro placement in very large-scale integration (VLSI) design. Experiments demonstrate that the proposed method significantly outperforms previous state-of-the-art (SOTA) methods.

All the reviewers agree that the investigated problem is important, the proposed approach provides a new research direction for macro placement problem, and experiments show good improvements. Overall, the paper is valuable for this community. The reviewers provided some constructive suggestions and asked clarifying questions. The authors have done a very good job in answering the questions during the rebuttal phase.

Overall, I recommend accepting the paper and strongly encourage the authors to revise the paper to reflect the reviewer-author discussion for the final submission.